# Environmental Impact of Textile Materials: Challenges in Fiber–Dye Chemistry and Implication of Microbial Biodegradation

**DOI:** 10.3390/polym17070871

**Published:** 2025-03-24

**Authors:** Arvind Negi

**Affiliations:** Faculty of Educational Science, University of Helsinki, 00014 Helsinki, Finland; arvindnegi2301@gmail.com or arvind.negi@helsinki.fi

**Keywords:** fiber chemistry, cellulose materials, synthetic polymers, reactive dyes, diazo dyes, synthetic dyes, colorants, bioremediation, microbes

## Abstract

Synthetic and natural fibers are widely used in the textile industry. Natural fibers include cellulose-based materials like cotton, and regenerated fibers like viscose as well as protein-based fibers such as silk and wool. Synthetic fibers, on the other hand, include PET and polyamides (like nylon). Due to significant differences in their chemistry, distinct dyeing processes are required, each generating specific waste. For example, cellulose fibers exhibit chemical inertness toward dyes, necessitating chemical auxiliaries that contribute to wastewater contamination, whereas synthetic fibers are a major source of non-biodegradable microplastic emissions. Addressing the environmental impact of fiber processing requires a deep molecular-level understanding to enable informed decision-making. This manuscript emphasizes potential solutions, particularly through the biodegradation of textile materials and related chemical waste, aligning with the United Nations Sustainable Development Goal 6, which promotes clean water and sanitation. For instance, cost-effective methods using enzymes or microbes can aid in processing the fibers and their associated dyeing solutions while also addressing textile wastewater, which contains high concentrations of unreacted dyes, salts, and other highly water-soluble pollutants. This paper covers different aspects of fiber chemistry, dyeing, degradation mechanisms, and the chemical waste produced by the textile industry, while highlighting microbial-based strategies for waste mitigation. The integration of microbes not only offers a solution for managing large volumes of textile waste but also paves the way for sustainable technologies.

## 1. Introduction

### 1.1. Fiber Composition and Its Significance

Fiber chemistry deals with the study of the composition, structure, and physiochemical properties of fibers. The chemical characteristics of fiber significantly influence its reactivity and are, therefore, considered in decision-making for scaling or developing efficient industrial processes. Fibers are generally characterized by their physical characteristics as long, thin strands of material that can be spun into yarn and then woven or knitted into fabrics. Depending on nature, fiber could be of natural or synthetic origin. For example, natural fiber, like cotton and wool, or synthetic fiber, like nylon and polyester. Based on their state (natural vs. synthetic), they exhibit specific physicochemical and rheological properties, indirectly or directly affecting their chemical properties and specifying their commercial applications. Some of their key mechanical and chemical features that are considered for their targeted applications are as follows:(a)Stretchability or elasticity, which is an ability of a fiber material to return to the original shape (dimensions) after stretching. Spandex is highly elastic, making it ideal for sportswear and stretchable fabrics; the unique properties of different fibers make them suitable for various applications. Self-healing fibers or shape-reforming fibers are the most valuable features in this category, and most synthetic fibers are known to possess this property.(b)Water retention or moisture absorbency: The properties of water retention or moisture within the fiber can be associated with natural fibers such as cotton and wool, which exhibit high moisture retention, thereby making them comfortable for clothing. Unlike cotton or wool, which do have functional groups in their structure that can make an H-bond with moisture or water molecules and thereby can retain moisture/water within the fiber, synthetic fibers are generally devoid of such functional groups, therefore, tend to exhibit lower moisture retention.(c)Strength (tensile strength vs. mechanical strength): The ability of fiber material to withstand tension without breaking. Nylon and polyester are known for their high tensile strength, while natural fibers typically exhibit lower ability in such properties.(d)Insulation or thermal properties: The ability of fiber to retain or dissipate heat. Wool is an excellent insulator, while polyester is often used in thermal wear due to its low thermal conductivity.(e)Resistance towards chemical penetration or average reactivity: This means the ability to withstand exposure to chemicals. Polyester and nylon are resistant to many chemicals, making them suitable for industrial applications. In contrast, due to the presence of an H-bond network in the structure of natural fibers, they tend to show lower reactivity towards certain chemicals, as further discussed in Section 2.(f)Molecular arrangement of fibers: The properties of fibers are largely determined by their molecular structure. Most fibers are polymers, which are long chains of repeating molecular units called monomers. The arrangement and chemical composition of these monomers affects their fiber’s characteristics, determining the suitability of fibers for specific applications.

### 1.2. Natural Fibers and Their Challenges

Natural fibers are biopolymers that can be derived from plants, animals, or minerals. Cellulose-based fibers are a type of plant-derived fiber. For example, cotton and linen are cellulose-based fibers composed of glucose units linked together by β-1,4-glycosidic bonds. Unlike other polysaccharides, such as dextran, arabinogalactan, amylopectin, and glycogen, which contain branched chains in their structures, cellulose is an unbranched polymer, meaning it lacks branching in the molecular structures. Additionally, cellulose is made of a single type of monosaccharide (that is, glucose), and is, therefore, classified as an example of a homopolymer. In contrast, some polysaccharides are made of different monosaccharide units; for example, chitosan consists of β-(1→4)-linked D-glucosamine (deacetylated unit) and N-acetyl-D-glucosamine (acetylated unit). Although these polysaccharides are made of similar units, subtle differences in the molecular arrangement led to distinctive physicochemical properties, which in turn provide them with various materialistic applications. In a broader sense, branched polysaccharides may possess a more crystalline nature as they exhibit a greater degree of hydrogen bond networking compared to linear polysaccharides, and, therefore, tend to be insoluble in water or aqueous solutions. The hydroxy groups in cellulose form hydrogen bonds, giving these fibers strength and the ability to retain/absorb moisture. Some common natural fibers include cotton which is a plant-based fiber composed mainly of cellulose, which is a complex carbohydrate. Cotton fibers are soft, breathable, and highly absorbent, making them ideal for clothing and other textile applications, while linen fibers are strong, moisture-absorbent, and dry faster than cotton. On the contrary, protein-based fibers, such as wool and silk, are composed of amino acids linked by peptide bonds. The specific sequence and structure of these amino acids determine the fiber’s properties. For example, the helical structure of keratin in wool provides elasticity, while the β-sheet structure of fibroin in silk contributes to its strength and luster. Wool is an animal-based fiber obtained from sheep. Wool fibers are composed of keratin (a protein). They are known for their warmth, elasticity, and moisture-wicking properties, while silk is produced by silkworms, and silk fibers are composed of fibroin (a protein). Silk is renowned for its smooth texture, luster, and strength.

### 1.3. Synthetic Fibers and Their Challenges

Synthetic fibers are man-made and typically made from petrochemicals. The properties of these fibers can be tailored by altering the chemical structure of the monomers and the polymerization process. For instance, the addition of aromatic rings in the polymer backbone of polyester increases its strength and thermal stability. Some common synthetic fibers include nylon, which is a polyamide fiber known for its strength, elasticity, and resistance to abrasion and chemicals, while polyester is a polymer made from terephthalic acid and ethylene glycol. Polyester fibers are strong, resistant to shrinking and stretching, and quick drying. However, acrylic fibers are made from polyacrylonitrile, are lightweight, soft, and warm, and often used as a wool substitute while spandex is also known as elastane, another type of synthetic fiber, which is made from polyurethane and is highly elastic, making it ideal for stretchable clothing.

## 2. Specifics of Color or Dyeing Chemistry Towards Fibers

### 2.1. Dye: Structure, Origin, and Classification

Dyes can be categorized into two main types based on their origin: natural and synthetic (human-made) dyes. As the names suggest, natural dyes come from nature, while synthetic dyes are created by humans. In 1854, Henry Perkin synthesized the first synthetic dye, Mauveine, from coal tar [1]. In recent years, the production and financial status of synthetic dyes have seen tremendous growth, reaching USD 17.56 billion in 2023, and projected to be worth approximately USD 28.75 billion by 2028. Synthetic dyes include types such as aniline, chrome, anionic, and cationic, and are applied in various categories like acid dyes, direct dyes, basic dyes, reactive dyes, disperse dyes, and vat dyes. These dyes have found commercial applications in industries such as textiles, food and beverages, paper, ink, and leather, forming a significant part of the value-added chain with extensive commercial coverage. One of the primary reasons for their widespread use in various commercial applications is their low cost and availability, which can be attributed to decades of industrial scaling. Their functional reactivity and structural versatility make them suitable not only for bulk industries like food, paint, textiles, and printing but also for specialized roles as excipients or thermoplastic agents in the fine chemical industry, including pharmaceuticals and cosmetics.

Unlike pigments, dyes are known for durable coloration properties, which can be achieved either by directly forming a bond with the materials or assisted through a chemical auxiliary (like mordant, salts, alkali, etc.). What sets them apart from pigment is also their solubility because dyes are water-soluble but pigments are not. However, in some cases, these terminologies are interchangeable; therefore, it is the responsibility of a color chemist or textile chemist to make sure that they are using the right terminologies. Chemically, dyes could be simple organic aromatic molecules or complexes where molecules are chelating or forming a complex with metal ions and must absorb light within the visible range of the electromagnetic spectrum, which is 400–700 nm. There is much debate about one definition based on the structural features of dye but as we are developing newer chemotypes we could now say that there must be a chromophore, which is an aromatic system or extensive conjugate system, while it could be possible that one or more auxochrome groups are part of that chromophore. Unlike chromophores, which are primarily responsible for color properties, auxochromes are actively involved in enhancing the color properties with shifting absorption spectra visible towards higher wavelengths or towards lower wavelengths; therefore, auxochromes are not directly involved in color production but they play a secondary role. Structurally, a chromophore can be a delocalized electron system with conjugated double bonds, for example, diazo groups (-N=N-), imine groups (-C=N-), nitro (-NO_2_), carbonyl (-C=O-) while an auxochrome could be electron donating groups (hydroxyl (-OH), amino (-NH_2_)) or electron-withdrawing (such as aldehyde (-CHO) carboxylic acid (COOH), sulfonic (SO_3_H) and methyl mercaptan groups (-SCH_3_)). There are numerous ways to classify synthetic dyes in textile industries, for example, based on fiber, chromophore type, dyeing, etc. However, the most common one is based on their ionic character, as shown in Table 1. In my opinion, the highest recorded structures are of diazo-based dyes and those that have sulfate groups; therefore, most of the dyes covered in an anionic dye category exhibit a high aqueous solubility. Azo dyes are known to make up nearly 60–70% of textile dyes, and are in the family of largest chemical colorants (such as mono, diazo, triazo, and polyazo dyes, depending on the availability of a number of azo bonds) [2], constituting nearly 70% of the annual production of synthetic dyes globally [3,4]. When looking at the azo dye structure it usually contains a diazo functionality (-N=N-), which could be connected to a symmetrical and/or asymmetrical aromatic system (while there are some exceptions as well).

### 2.2. Dyeing Chemistry and Challenges

#### 2.2.1. Dyeing of Natural Fibers

##### Cellulose Fiber Dyeing: High Usage of Alkali, Salts and Reactive Dyes

The cellulose fiber industry faces significant challenges due to the chemistries involved in processing. Being the most abundant polymer in nature, it still lacks wide acceptability in commercial industries due to its chemical inertness, which can be correlated to the presence of the hydrogen bonding network and crystallinity in its molecular structure. Just to put that into perspective with synthetic fibers, cellulose-based fibers require a higher dye concentration per liter because dye exhaustion onto the native cellulose fibers is generally poor. Therefore, to improve its reactivity or enhance the chemical penetration, cellulose-based fibers are generally subjected to preactivation before undergoing any application-targeted processing or dyeing. For example, the commercial reactive dyeing of cellulose fibers (materials) is typically carried out in an aqueous medium, requiring the addition of electrolytes (for example, alkali metal chlorides or alkali metal sulfate) to enhance dye exhaustion due to the poor substantivity of reactive dyes towards cellulosic fibers.

Due to inherited flaws in the reactive dyes that are used for commercial reactive dyeing of cellulose fibers, they contain multiple sulfate groups that enhance their aqueous solubility. As a result, it is a common practice to add salt to the dyeing bath to induce a salting-out effect. This effect facilitates the migration of the dissolved dye (in water) towards the cellulose fiber. Therefore, a significant amount of electrolytes (salts) is typically added during cellulose fiber reactive dyeing at a commercial level, ranging from 30 to 100 g/L. This addition of electrolytes (salts) undoubtedly enhances the exhaustion of dyes and can be a key factor in producing low-to-high color shade or color depth, but they may also impact the final hue or uniformity of the dyeing.

Following this, an alkali (such as Na_2_CO_3_ or NaOH) is added to promote the fixation of the dye molecules onto the fibers. Chemically, because of the alkali addition, this facilitates the covalent bonding between the hydroxy groups (comparatively with primary hydroxy groups refers to C_6_-*OH* than those secondary hydroxy groups in the form of C_2_-*OH* or C_3_-*OH*) of cellulose and the dye molecules. The bonding primarily occurs through electrophilic reactions, typically at the most electrophilic atom position of the dye molecule, which is usually the chloro position of a monochlorotriazine or the vinyl terminal of a vinyl sulfone group as the reactive dye.

Cellulose fibers possess limited reactivity; therefore, dyes need to have at least one reactive functionality or group to form covalent bonds. The secondary purpose of adding a high amount of salt in cellulose fiber dyeing is, to an extent, that it keeps the cellulose in the ionic form as soda cellulose (soda cellulose only in case sodium chloride is used). Consequently, the addition of alkali salts in the form of carbonates (Na_2_CO_3_, K_2_CO_3_, Cs_2_CO_3_) or hydroxide (NaOH, KOH, LiOH) brings more deprotonation of the hydroxy groups of the glucose units of cellulose fibers, making them available to form covalent bonds with reactive dye molecules (mostly at that atom or position onto the reactive dye structure which is highly electrophilic or electron-deficient). Such a reaction between a reactive part of a reactive dye with a hydroxy group of cellulose is called dye fixation on the fiber. Therefore, to enhance the dyeability or reactivity of the cellulose fibers, preprocessing with alkaline solutions is common in practice, which breaks these intermolecular hydrogen bonds of cellulose microfibrils, as well as using deprotonation, to an extent, for some of the hydroxy groups, ultimately increasing the availability of the free hydroxy groups at the C2, C3, and C6 positions of glucose for a dye fixation reaction, as shown in Figure 1. Some of these OH groups become nucleophilic, particularly the C6 hydroxy group, due to its primary alcohol nature. This nucleophilicity, mainly from the C6 hydroxy group, allows it to react with the reactive part of reactive dyes (such as monochlorotriazine or vinyl sulphonyl groups), forming covalent bonds with the OH groups of the glucose units in cellulose during dyeing; however, a small proportion of reactive dye that can be deposited on the fiber surface can be washed off easily. Although using natural extracts for cotton dyeing could provide a sustainable solution [1,2,3], natural pigments lack reactive groups, meaning there is no fixation of these pigments or colorants.

At the molecular level, the chains within cellulose fibers possess a hydrogen bonding network, making it unreactive and less chemically penetrable. Therefore, most cellulose fiber processing requires chemical auxiliaries, as well as dyes that contain at least one reactive group to form covalent bonds with hydroxy groups. The inclusion of reactive groups in the structure of reactive dyes (such as triphenyl triazines) indeed provides reasonable fastness properties to their dyed cellulose fibers, but these are also known to hydrolyze in the water. Therefore, some of the percentages of reactive dyes remain unreactive and, ultimately discarded in wastewater. Another issue of cellulose fiber dyeing with such reactive dyes is that these dyes are challenging to synthesize; therefore, commercial reactive dyes that are used at industrial scales are limited to a few reactive groups, such as monochlorotriazine, dichloro triazine, and vinyl sulphone. Based on the presence and nature of reactive groups, these reactive dyes can be categorized as bifunctional or monofunctional. For example, if there are two reactive groups available in one dye structure then it will be called bifunctional, if it has only one then it will be called a monofunctional reactive dye. However, if there are two different reactive groups then it can be called heterobifunctional (Reactive Red 239, RR239), while if both reactive groups are the same then it is called homobifunctional, as shown in Figure 2. Although monochlorotriazine is the most common reactive group found in commercial dyes, other reactive groups are also can be found, such as vinyl sulfonyl groups, pyridinium chloride, acryl groups, dichlorotriazine, monochloro-hydroxytriazine, and various heterocyclic rings. Reactive dyes usually show a high electrophilicity due to the presence of a high electronegative atom or substructure in the molecular structure (in the form of a quaternary ion or halide), therefore, they can easily undergo a nucleophilic attack. For example, looking at the monochlorotriazine reactive group, which has a monosubstituted chloro group on an electron-deficient triazine ring, this chloro-substituent of monochlorotriazine provides the location for an upcoming nucleophilic attack from the cellulose hydroxy groups to form a covalent bond, which is non-ionic in nature. However, due to the presence of such high electronegative groups (such as monochlorotriazine) in reactive dyes, they are exclusively used for cellulose dyeing. The dyeing of cellulose fibers typically occurs in an alkaline medium since reactive dyes are highly sensitive to pH changes, which can lead to hydrolysis, transforming them into less reactive or water-soluble forms. As a result, this reduces their effectiveness and generates significant dye waste, contributing to pollution in waterways [4,5].

##### Protein Fiber Dyeing: Application of Mordants for Wool Dyeing

Protein fibers, such as wool and silk, are composed of amino acids and are produced by animals. These fibers typically have an amide peptide backbone. Due to the presence of ionic groups in these protein fibers, they are usually dyed with acid or mordant dyes. Acid dyes are preferred over basic dyes primarily because they typically provide better lightfastness properties. These dyes mainly consist of anionic groups, with sulfate groups being the most common, although some dyes also contain carboxylic groups. Acidic dyes feature a chromophore, often characterized by multiple sulfate groups. The sulfate groups (sodium sulfonate salts) in the dye structure provide ionic characteristics, enhancing aqueous solubility. In acidic conditions, the NH_2_ groups on protein fibers become protonated, attracting the negatively charged acidic dyes with sulfate groups. Other forces, such as Van der Waals, dipolar interactions, and hydrogen bonding, also aid in the dyeing of protein fibers. Substitution with hydroxy, carboxy, or any other hydrogen bond donor group at the ortho position relative to the diazo bond of acidic dye can form intramolecular hydrogen bonds. Such structural features improve resistance to washing in alkaline conditions and enhance lightfastness. To improve dyeing behavior, researchers often use mordants, which are typically metal salts (such as aluminum or iron salts) or bio-based mordants (such as tannic acids). Mordant dyes generally involve complexation reactions between inorganic metals (derived from mordants) and organic ligands (usually derived from dyes, especially the chromophore part of a dye molecule), enhancing color shades and fastness properties. Acid dyes are named after their dyeing processing, where an acidic pH is used and is primarily applied to protein fibers such as wool. However, they show application beyond wool dyeing, especially in food industries, and as theragnostic agents for staining organelles [6,7,8,9]. Structurally, acid dyes contain anionic groups that provide high aqueous solubility, making it challenging for the dye molecules to migrate from the dye solution toward the fiber surface. To address this, their aqueous solubility is often reduced, so that these dissolved molecules are encouraged to migrate toward the fiber. In textile applications, acid dyes are used in acidic dye baths, where they interact with the fibers that have amide bonds (such as protein fibers, like wool, and silk), or synthetic fiber (polyamide fibers: nylon) through ionic bonds. These ionic bonds (often, H-bonds) are formed between the NH_2_ groups (protonated, which is achieved by lowering the pH by adding acid during dyeing) of the fiber and the acidic groups (SO_3_H and COOH) of the dye. Generally, acid dyes exhibit poor wash fastness but better lightfastness. The opposite electrical charges of the dye and fiber result in a faster uptake of the dye (exhaustion) and, even during processing, higher concentrations of electrolytes are added to slow down dye uptake so that even shades can be achieved. The addition of acid or a lower pH results in the protonation of the amino groups present in the fiber, which produces a positive surface charge (cationizing the fiber). Meanwhile, an increase in the temperature enhances the ionic interactions between the positively charged protein fibers and negatively charged anionic dyes.

There are a number of parameters that could affect the extent and efficiency of the acid dyeing of fibers, while some account for a large impact, such as the pH used during the dyeing process, nature of the fiber, dye concentration, liquor–-dye ratio, temperature, dye molecule ability to migrate, and fastness properties. Dyeing with acid dyes utilizes different bonding mechanisms to interact with the fibers; however, in my opinion, the involvement of hydrogen bonding and ionic bonding are the primary mechanisms, while reports indicate the involvement of Van der Waals forces and hydrophobic interactions. Importantly, the structure of acid dyes plays an important role in their dyeing; for example, a few acid dyes are known to achieve reasonable dyeing in water, while others (most acid dyes) become activated upon lowering the pH of the solution to achieve efficient dyeing. During acid dyeing of protein fibers (mainly those that are derived from animal sources) and synthetic fibers (such as polyamide, like nylon fibers), there is generally an addition of acid, which lowers the pH of the dyeing bath; therefore, these fibers usually become protonated generating numerous cationic sites that have a positive charge to interact the anionic groups of acid dyes (mostly negatively charged sulfate groups). Achieving reasonable fastness properties exhibits the strength of high participation of ionic bond interactions in acid dyeing.

In general, synthetic fibers are known to achieve better dyeing results compared to natural fibers. However, the dyeing chemistry of synthetic fibers presents unique challenges that differ from those associated with natural fibers. One significant issue is microfiber pollution, which is covered in another section of this manuscript.

### 2.3. Strategies to Improve Dyeing or Coloration Processes

Most dyeing issues arise with natural-based fibers, particularly cellulose-based fibers. Since cellulose-based fibers have a higher production turnover annually at the global level, most strategies have been developed for these fibers or their dyeing chemistry. This includes methods such as blending with other fibers and using the dye chemistry of the blended fibers, such as polycotton, which is a blend of polyester and cotton in different compositions. Additionally, cationization has gained significant interest among researchers as a part of dyeing chemistry.

#### 2.3.1. Blending Fiber or Composites

Blending fibers or developing new composite materials has shown significant promise in enhancing the mechanical, thermal, and conductive properties of fibers [10,11,12].

These innovations have led to the creation of stronger, more durable materials with improved heat resistance and, in some cases, better electrical conductivity [13,14,15,16,17,18].

Furthermore, modifying the structure of fibers through blending can enhance their surface characteristics, such as roughness and chemical reactivity. These improvements often result in better dye absorption and color retention during the dyeing process [19,20,21]. Numerous studies have shown that such structural modifications can enhance the dyeing performance of specific fibers, leading to better color fastness and uniformity.

Consequently, the cellulose processing or dyeing industry relies on a supply chain consisting of several sequential steps, with coloration being a crucial one that significantly impacts the commercial cost of the final product. Researchers have attempted to blend cellulose fibers with other fibers to enhance their dyeability. However, due to physicochemical differences among the fibers, the focus has mainly been on using colorants for the blended fibers, reducing the value of blended cellulose fibers in mixed compositions. The application of reactive dyes in the 1950s for cellulose material coloration (dyeing) was identified, where the reactive dyes in an alkaline pH were found to form non-ionic covalent bonds with the hydroxy groups of glucose units (cellulose fibers). However, they were successfully commercialized due to their outstanding fastness properties for cellulose-based fibers. Due to their high chemical reactivity, they are highly prone to hydrolysis and degradation during their industrial processing, which is further facilitated by the alkaline pH, which is a common parameter required during their dyeing, and, therefore, exhibit a high chemical and environmental footprint. To address this, there are some solutions proposed and investigated:(a)Synthesizing reactive dyes that have two or more reactive groups (as can be called bifunctional or polyfunctional reactive dyes) offers more opportunity for hydroxy groups of cellulose to react with these dyes and, therefore, improve the overall dyeing process; however, such innovations have limited commercial success as this will add burden on synthetic chemist to design a molecule using multistep synthesis, and also increase overall cost. Therefore, only a handful of bifunctional reactive dyes are known to date.(b)Using greener or more environment-friendly solvents, such as alcohols to decrease dye hydrolysis as it is well known that the alkaline solution (where water can act as a nucleophile to complete with the hydroxy group of cellulose) leads to the hydrolysis of the reactive dye. Therefore, incorporating greener solvents could lower the rate of hydrolysis of reactive dye during their commercial processing. However, in comparison, it is commercially not feasible for textile industries to use solvents as a replacement for water.(c)The inertness of cellulose fibers and the use of highly reactive dyes and chemical auxiliaries, ultimately increase the chemical waste in the wastewater produced from the textile industries, necessitating the development of cellulose blends with other fibers. Blending of fibers allows the dyeing of the blended fibers rather than just cellulose. Polycotton, as an example, is a blend of polyester and cotton and has been commercially successful. Although the dyeing of polycotton has been optimized and shows improved dye absorption, the presence of polyester can complicate the dyeing chemistry. This complication arises because polyester can inhibit the reactivity of cellulose fibers, thereby reducing their effectiveness in dye applications. One argument for the decreased societal and economic importance of dyeing the cellulose part of polyester fibers is the low reactivity of cellulose toward specialized dyes. As a result, achieving broader commercial success in this area is currently limited. However, in my opinion, the process will mature over time, and, therefore, the cost of industrial processing will also decrease. However, the use of blended cellulose fibers was intended not for dyeing the cellulose part but rather for dyeing the polyester fibers. This intention may pose challenges for the industry, as it can lead to a perceived value loss of one of the components of the material.

#### 2.3.2. Sustainable Fiber Technologies

The textile industry is undergoing a significant transformation driven by the need for sustainability. Traditional textile production has been associated with substantial environmental impacts, including high water usage, chemical pollution, and reliance on non-renewable resources. However, advancements in sustainable fiber technologies are paving the way for a more eco-friendly future. There are several sustainable fiber technologies available for textiles, which are still evolving. In my opinion, these technologies primarily focus on chemical processing. Most cellulose fibers are either produced using methods with a lower chemical footprint or are chemically modified to enhance their textile applicability. These innovations in textile production represent a positive shift toward more sustainable materials. Covering a comprehensive description of all current technologies is beyond the scope of this review; therefore, readers are encouraged to read the following papers or titles for more in-depth information: Impact of recovery and recycling in textile industries [22]; Technological and socio-economical aspects of natural fibers [23]; Advanced sustainable woven natural fiber polymer composites [24]; Sustainable reuse of waste tire textile fibers [25]; Impact of technical textiles and synthetic nanofibers on environmental pollution [26]; and Sustainable solutions and textile Innovations using biopolymers from plastic industries [27]. However, from a broader perspective, there are many other publications that appeal to a wide range of readers, such as Digital technologies in the sustainable design and development of textiles [28,29]; Consumer attitudes towards textile fiber origin [29]; The impact of digital technologies on circular transformation in textiles [30]; and The enhancement of textiles with digital technology investments [31]. There are also digital advancements in sustainable apparel, with lessons learned from textile development projects [32], and the progression of sustainable wearables using innovative textile techniques [33]. Additionally, several research books and book chapters may interest readers [34,35,36]. However, below are the key highlights of major processes and technologies that can be effectively applied to promote sustainability in commercial textiles.

(a)Renewable and Bio-based fibers

One of the most significant developments in textile production is the rise of bio-based fibers, which are produced using efficient processes that have a lower chemical and environmental footprint. These fibers are sourced from renewable resources, such as plants, algae, and other materials, helping to reduce dependence on fossil fuels and therefore tend to have lesser environmental impact. Here are some notable examples of sustainable fibers and technologies, each highlighting its unique environmental benefits:

Ioncell: Ioncell technology is an innovative and sustainable method for producing high-quality textile fibers from cellulosic materials. Developed by researchers at Aalto University and the University of Helsinki, this technology utilizes a non-toxic ionic liquid to dissolve cellulose, which is then transformed into strong, durable fibers through a dry-jet wet-spinning process. A significant contribution to developing this technology comes from Professor Herbert Sixta, who has published important research on the subject [37]. The entire process is heavily influenced by the ionic liquid used. Ionic liquids (ILs) are salts that remain in a liquid state at or below 100 °C, which grants them unique physicochemical characteristics and provides flexibility in the experimental design of sustainable fiber production. For instance, ionic liquids exhibit excellent thermal stability, often withstanding temperatures of up to 250 °C for prolonged periods.

Cellulosic materials typically have poor solubility in organic solvents and even in water, as it is highly dependent on the presence of degree of crystallinity. However, polar ionic liquids display versatile solvent characteristics that enable the dissolution of cellulose materials. Additionally, ionic liquids exhibit high ionic conductivity and low volatility, contributing to efficient experimental designs that can be fine-tuned for specific applications. Key features of Ioncell technology include the ability to recycle textile waste into new fibers, as well as the production of traditional textile fibers such as cotton, viscose, and polyester.

The crystallinity of cellulose is due to a complex network of hydrogen bonds (H-bonds) formed between the hydroxy groups of glucose units, both within the same chain and between different chains [38]. These intermolecular and intramolecular hydrogen bonds give cellulose crystals their strength and toughness, resulting in favorable mechanical properties. Additionally, these bonds make cellulose insoluble in water and most organic solvents. This insolubility limits its reactivity and processability for various applications. As a result, researchers have focused on the challenge of dissolving cellulose to enhance its usability in different processing and spinning operations [39,40]. Before the development of ionic liquids, extensive research was conducted on alternative cellulose solvents and their associated limitations. The Ioncell concept is further explored using different experimental setups [41,42,43,44].

Tencel (Lyocell): Lyocell, commonly known by the trademark Tencel, is a semi-synthetic fiber derived from wood pulp. Researchers aimed to develop a type of fiber similar to rayon but using a less chemical-intensive process than the traditional viscose method, which ultimately led to the creation of lyocell. In contrast to rayon production, lyocell fabrication does not involve carbon disulfide, making it a safer option for both workers and the environment. Carbon disulfide is recognized for its neurotoxic effects, and the viscose process generates substantial amounts of contaminated wastewater. In response to these issues, lyocell technology has emerged as a sustainable alternative. The U.S. Federal Trade Commission defines lyocell as “*a fiber composed of cellulose precipitated from an organic solution in which no substitution of the hydroxy groups takes place, and no chemical intermediates are formed*”. This definition highlights the eco-friendly nature of lyocell production compared to the viscose process.

Lyocell is known for its soft texture and various organoleptic properties when compared to other fibers. Importantly, its production employs a closed-loop method, allowing for the effective recycling of solvents and water, which significantly reduces waste. Rayon production involves dissolving cellulose (and using carbon disulfide) to create cellulose xanthate, which is then regenerated into cellulose fibers, leading to the development of various solubilization methods. Traditionally, the cuprammonium process, which uses an ammonium solution with copper salts, was one of the initial methods and primarily used for producing fibers like cupro, which are similar to rayon. Importantly, the cuprammonium process is different from rayon production but is also part of the cellulose fiber production family. In contrast, other methods such as the viscose method, which is commonly utilized on an industrial scale, involve pretreating cellulose with a strong alkali and then treating it with carbon disulfide to produce a cellulosic xanthate, which is subsequently converted back into cellulose fiber. Viscose can be produced from wood cellulose, but other lignin-free materials can also be utilized. Recently, the invention of enzymes and photocatalysis with lignocellulosic materials has achieved more sustainable pulping methods [45,46,47,48,49]. One significant advancement developed and scaled at the industrial level is the lyocell method, which uses N-methyl morpholine N-oxide (NMMO) and employs dry-jet wet spinning.

NMMO dissolves lignocellulosic materials (such as wood pulp) that have a high cellulose content with minimal lignin and hemicellulose. This process utilizes a direct solvent rather than an indirect dissolution method like the xanthation-regeneration route used in viscose production. This means, NMMO directly dissolves cellulose without needing to form an intermediate derivative, unlike the viscose process, which involves creating a xanthate intermediate before regeneration. Notably, NMMO is recycled with over 95% recovery efficiency. Like any cellulose fiber, lyocell is composed of long chains of glucose units. The alignment of these chains provides high crystallinity, resulting in its favorable mechanical properties. The cellulose in lyocell fibers typically exists in the form of cellulose II, a polymorph that is more stable and less prone to swelling compared to cellulose I, which is found in natural fibers like cotton. Within the fibers, the cellulose chains are organized into microfibrils—bundles of cellulose molecules embedded in an amorphous matrix—providing flexibility and enhancing the fiber’s mechanical properties. The smooth and uniform surface of lyocell fibers contributes to their soft feel and high moisture absorption capacity, making them an excellent choice for textiles that require comfort and breathability. Lyocell can be blended with other fibers, including cotton, silk, rayon, nylon, wool, linen, and polyester. It shares many properties with other fibers such as cotton, linen, silk, ramie, hemp, and viscose rayon, with which it is chemically closely related. Lyocell is 50% more absorbent than cotton and has a longer wicking distance compared to modal fabrics of a similar weave.

(b)Waterless Dyeing Technologies

Conventional dyeing processes are known for their high consumption of large volumes of water, leading to significant wastewater pollution. One major advantage of waterless dyeing technology is that it promotes a closed-loop system, which enhances sustainability by minimizing the loss of solvents and dye baths. Key examples include supercritical carbon dioxide dyeing [50,51,52,53,54,55,56,57,58], surface plasma-assisted dyeing [59,60,61,62,63,64,65,66,67,68,69,70], atomic layer deposition [71,72,73], and solvent-based dyeing [74,75,76,77,78,79,80,81,82,83]. However, a solvent other than the aqueous phase has been well explored, such as eutectic solvents [84,85,86,87,88]. However, for dyeing, the medium volume requirement is so high; therefore, non-aqueous solvent-based dyeing is often considered more expensive than aqueous base dyeing even though it is a much more sustainable process. Therefore, we require more innovation in sustainable fiber technologies that can reshape the textile industry, offering eco-friendly alternatives to traditional materials and processes.

#### 2.3.3. Cationization of Fibers

There are various cationizing agents available for different types of fibers, which can come from different sources. For example, both biobased and synthetic cationizing agents have demonstrated significant improvements in the surface reactivity of natural fibers after application. Various reports indicate that chitosan, the second most common natural biopolymer after cellulose, can be used as a biobased cationizing agent for cellulose-based materials [89]. Chitosan and cellulose are both polysaccharides, which means they can exhibit hydrogen bond interactions with each other when present in a composite form. Additionally, they can be covalently linked together using cross-linking agents. One such agent is cyanuric chloride, which can facilitate the formation of covalent bonds between chitosan and cellulose materials. There is substantial evidence that chitosan acts as a natural cationizing agent, which is beneficial for reactive dyeing as well as for direct dyes. For example, a pad–dry–cure process using chitosan nanoparticle-treated cotton has demonstrated improved dyeability with acid dyes, where Acid Red 88 exhibited a higher K/S value compared to Acid Blue 317 [90]. While, in my opinion, this discrepancy (Acid Red 88 exhibits a higher K/S value compared to Acid Blue 317) may be related to the solubility of Acid Blue 317. The Acid Blue 317 dye structure reportedly includes a metal ion that probably forms a coordinate bond with the chromophore. This suggests that the presence of positively charged metal ions could contribute to ionic repulsion from the positively charged surface of chitosan nanoparticles when applied to cellulose fabric. Additionally, the quaternary centers present in chitosan possess intrinsic antibacterial properties and can also function as scaffolds for delivery agents [91,92,93,94,95,96]. One of the key synthetic cationizing cellulose fiber agents that is used is 3-chloro-2-hydroxypropyl trimethylammonium chloride (CHPTAC). The CHPTAC is predominantly used for cellulose fibers due to its reactive chemistry towards cellulose fiber; therefore, it forms an in situ epoxide which is reactive enough to form a covalent bond with the incoming nucleophile (hydroxy groups of cellulose). As a result, it permanently cationizes cellulose fibers, improving their ability to interact with dyes and, therefore, enhancing the exhaustion of anionic dye onto the fiber. However, in situ epoxide formation requires a reaction of CHPTAC with NaOH, which raises the pH of the solution above 12.0. In my opinion, this increase in pH further enhances the fiber nucleophilicity as cellulose OH tends to deprotonate in alkaline pH as shown in Figure 3. To validate this point, a control experiment should be conducted in which CHPTAC is added to a dye bath containing cellulose fibers but without NaOH.

Furthermore, it seems that CHPTAC consumes some of the hydroxy groups for cellulose fibers, therefore, one could argue that there are not enough remaining hydroxy groups after cationization for dyeing. In my opinion, it could be a trade-off where some of the OH groups were preoccupied with the cationizing process, assisting in the migration of dye molecules toward the fiber due to the positive charge generated on the fiber surface. Another argument is that cationization promotes a high ionic interaction that resembles those of ionic salts (that are traditionally known to be used in cellulose fiber dyeing). Thus, a techno-economic evaluation comparing the performance of these methods would be necessary. Additionally, the number of negatively charged groups in the dye structure is crucial for achieving optimal dye exhaustion on the fibers. While some studies have compared the number of sulfate groups critical for dyeing performance, the use of dyes with varying molecular weights leads to ambiguous conclusions. This highlights a gap in dyeing chemistry that requires further investigation. Dyes containing sodium sulfonate groups of anionic nature, such as Acid, Direct, and Reactive dyes, tend to exhibit improved dyeing results when applied to precationized cellulose fibers.

#### 2.3.4. Salting-In/Salting-Out

The dyeing of natural and synthetic fibers typically requires a certain amount of electrolyte concentration to assist the dye molecules to migrate from the highly soluble aqueous solution toward the fibers. Unlike synthetic fibers, which are generally known for their good dyeability, natural fibers are recognized for their poor dyeability (in comparison), requiring the use of high electrolyte concentrations for dyeing. For example, reactive dyeing of cellulose fibers is considered less environment friendly, producing large volumes of chemical waste, including electrolytes, unfixed or hydrolyzed dyes, and alkaline solutions, which are known to produce ecotoxicity. Even under optimal conditions, the fixation rate of reactive dyes that have monochlorotriazine exhibits 55–70%, while bifunctional reactive dyes (of homo or hetero nature) achieve a fixation rate ranging from 70 to 85%. Treating the resulting effluent, which contains high salt and alkaline levels, presents significant technological and economic challenges. To address these issues, researchers have explored numerous approaches to enhance cellulose reactivity and fiber–dye chemistries, ultimately reducing salt consumption, and enhancing dye substantivity. These strategies involve using homo and hetero-bifunctional reactive dyes and cationizing cellulosic materials to improve dyeing efficiency. However, despite those inventions, their scalability is still debatable, and, therefore, conventional dyeing still faces sustainability challenges. Interestingly, one approach to replacing salts or electrolytes in the dyeing process is known as salt-free dyeing, which involves using multiple solvents. Some reported binary solvent mixtures, primarily hydroalcoholic solutions that combine water with various types of alcohols, have been found to enhance the solubility of certain dyes. As a result, these methods require a comparatively lesser amount of chemical auxiliaries, particularly salts, than traditional water-based dyeing processes.

Historically, solvent dyeing has been commonly practiced for synthetic fibers since the 1970s, as perchloroethylene was used for their dyeing solvent, while supercritical carbon dioxide fluid introduced in the 1990s, offers a possible solution to mitigate water pollution and environmental toxicity associated with synthetic fiber dyeing. However, research on cellulose-based fiber-reactive dyeing using organic solvents or solvents other than water is limited. Most studies have focused on binary solvent systems, short-chain alcohols, supercritical CO_2_ co-solvent systems, and polar protic solvents.

The alkali pretreatments are essential for achieving effective reactive dyeing of cellulose fiber in non-aqueous solvents, as they enhance dye substantivity within the fiber structure. The use of binary solvents introduces the concept of co-solvency, a phenomenon where two solvents work together to improve the solubility of the dye. Typically, anionic or reactive dyes contain sulfate groups that contribute to their high solubility in water. However, the chromophores of these dyes are organic in nature and exhibit limited to no solubility in aqueous solutions. As a result, these chromophores (present as a substructure in the dye) tend to dissolve more effectively in organic solvents than in aqueous mediums. However, this concept faces challenges from the perspective of commercial scalability. Therefore, mixing organic solvents with water can improve solubility but also affect other physical properties such as boiling point and vapor pressure. An increased boiling point of the dyeing solution solvent system poses difficulties in recycling or recovering these solvents during repeated dyeing cycles. As a result, the commercial use of mixed solvents is often debated, making them less suitable for promoting sustainability in the dyeing process. Consequently, mono-solvent systems are primarily employed in commercial dyeing practices. Periyasamy et al. utilized an alcoholic solution instead of an aqueous solution for dyeing and replaced the conventional alkali-salt solution (composed of Na_2_CO_3_/NaCl or NaOH/NaCl) with organic alkoxides [4], as illustrated in Figure 4. In their study, they compared the advantages of using organic alkoxide-based alcoholic solutions for dyeing with those of inorganic alkali-salt-based aqueous dyeing.

#### 2.3.5. Thermolinkable and Photolinkable Dyes

There are certainly high concerns with the commercial processing of most natural fibers, as their process requires the addition of chemical auxiliaries to fix the dye onto the surfaces of these fiber materials. This has led to interest in developing dyes that can be fixed using heat or light. Dyes that can be fixed by heat are known as thermolinkable or thermoactive dyes, while those that require light for fixation are referred to as photolinkable or photoactive dyes. It is important to differentiate photochromic and thermochromic dyes, from photolinkable and thermolinkable dyes. The former (photochromic and thermochromic dyes) change color in response to light and temperature, respectively [97,98,99,100,101,102]. Consequently, their applications differ significantly. Some well-known examples of these types of dyes can be found in smart textile materials.

Photolinkage dyes are known for their ability to activate themselves or enhance their dyeing/coloration properties/reactivity in the presence of light. Most photolinkage dyes are based on spironolactone structures, whereas cellulose dyes are classified as reactive dyes, meaning they require a reactive group in their structure to fix onto cellulose fibers. Therefore, developing a photolinkage dye with a reactive group suitable for cellulose dyeing requires a complex multi-step synthesis process. This multi-step synthesis not only complicates the commercial production of photolinkage-based reactive dyes but also increases the overall processing cost. Therefore, it is one of the major reasons why we do not see enough photolinkable reactive dyes in development for textile applications.

However, photolinkable dyes could be used for synthetic fibers as they do not require a reactive part in their structure and, therefore, do not require multi-step synthesis for their production. On the contrary, thermolinkable dyes are reported with many more examples, as they tend to activate at a high temperature. For example, the triazine core of reactive dye that is used for cellulose fiber is reported with pyridinium chloride or nicotinic acid (also known as “vitamin B3”) exhibiting reactivity towards cellulose fiber and, therefore, attain reasonable dyeing at room temperature. However, the results are limited to the triazine core, which demonstrates that “triazine” itself is significantly important to achieving thermolinkable reactive dyeing. However, thermolinkable dyes could reduce the usage of chemical auxiliaries that are used in high concentrations (alkali and salts) for cellulose dyeing or functionalization.

Negi et al. proposed a symmetrical reactive molecule featuring thermolinkable groups utilizing a triazine core, specifically a 2,4,6-trisubstituted triazine for cellulose activation [5]. Unlike other trisubstituted aromatic ring structures [103,104], these trisubstituted triazine functionalities create a connection/link between cellulose and functionalizing agents, as illustrated in Figure 5. Importantly, these trisubstituted triazines operate in two steps to accomplish cellulose functionalization. Their high reactivity, based on leaving group chemistry, allows them to readily react with the hydroxy groups of cellulose fibers or other cellulosic forms. In this reaction, one of the three leaving group positions symmetrically is utilized for cellulose activation, while the remaining two reactive positions on the triazine core are still available for tethering/bonding/fixing with functionalizing agents that contain specific nucleophilic groups, such as -SH, -OH, -NH_2_, or -OPh.

Importantly, most natural compounds, especially those with polyphenolic structures, and a few synthetic chromophores that contain both phenolic and amino groups within their structure, could exploit this strategy for their fixation onto the cellulose fiber surface or for dyeing applications.

## 3. Environmental Impact of Textile Fibers and Dyeing Process

### 3.1. Microplastics from Textiles: Environmental Toxicity and Ecological Consequences

Microplastics from textiles: Policies, regulation, and Estimations: Despite the growing awareness of microplastic pollution and its impact on the ecosystem, synthetic fibers from the textile industry remain a significant contributor to this problem. However, the demand for synthetic fibers in the textile sector remains high. The wearing and washing of textiles made from synthetic fibers generate a significant amount of microplastics in the environment. European Topic Centre on Circular Economy and Resource Use (ETC/CE) published a report in 2022/23 that highlights several key points:It is estimated that over 14 million tons of microplastics have accumulated on the ocean floor, damaging ecosystems and impacting wildlife and humans.Synthetic textiles contribute nearly 8% of the microplastics released into oceans in Europe. However, worldwide, this figure varies for textiles, ranging from 16% to 35%. This results in nearly 200,000 tons of microplastics entering marine ecosystems annually.Most microplastics from textiles are released during their initial washings. Fast fashion materials, which are used for only short periods, tend to wear out quickly due to their inferior quality, leading to particularly high levels of microplastic release.The release of microplastics from textile materials could potentially be reduced through sustainable manufacturing practices and improved end-of-life processing and disposal management.

Throughout the lifecycle of textiles, they continuously contribute to microplastic pollution, primarily by shedding microfibers. These microfibers can originate from both synthetic and natural fibers, while other components and accessories, such as buttons, prints, coatings, and glitter, are known to produce additional microplastics.

Some studies suggest that the annual accumulation of microplastics in the oceans from synthetic textiles ranges from 0.2 to 0.5 million tons. Other research indicates that approximately 35% of the microplastics accumulated in the oceans come from synthetic textile washings, while the United Nations Environment Program (UNEP) estimates this figure to be around 16%. Importantly, approximately 13,000 tons of textile microfibers, equivalent to nearly 25 g per person, are released into surface waters each year in Europe. This accounts for 8% of the total primary microplastic releases into water, as most homes are connected to wastewater and sewage treatment systems.

Research has largely focused on the release of microfibers during the washing of synthetic textiles, with wastewater being the primary route for entering aquatic environments. As illustrated in Figure 6, during the lifecycle of textile fibers, microfibers are produced at various stages, including manufacturing, wearing, and washing, leading to their dispersal in water, soil, and air.

Interestingly, it was initially believed that successive washes would reduce microfiber release. However, as textiles age, their intermolecular strength decreases, which can actually increase the overall microfiber release in some cases. Furthermore, the growing trend of fast fashion significantly increases the amount of microfibers released, as these low-quality materials wear off easily. Fast fashion products also tend to stay on trend for a short time, resulting in substantial textile waste. This poses a serious challenge to achieving sustainable goals and developing a circular economy, as most of these materials are wasted, harming the environment. Both industrial and home washing of textiles play critical roles in microfiber release. Some industries have opted for additional treatments in their washing processes to filter out some of the microfibers. In contrast, washing at home mainly occurs in washing machines and, to a lesser extent, through hand washing. Therefore, washing machines equipped with advanced filters or treatment systems can help minimize microfiber release. Moreover, research suggests that washing parameters significantly influence microfiber shedding. Preliminary findings indicate that longer washing cycles increase wear and tear and that elevated temperatures can damage the fiber structure, leading to increased microfiber release. Additionally, the abrasiveness of particles from washing powders can enhance wear and tear, resulting in more shedding compared to liquid detergents. In contrast, using fabric softeners may decrease abrasiveness and friction, thereby reducing fiber shedding. There has also been some debate regarding the effectiveness of top-loading versus front-loading washing machines. Evidence suggests that top-loading washing machines tend to cause significantly more shedding due to greater abrasion during the tumbling process compared to front-loading machines.

There is an ongoing debate among policymakers and environmentalists regarding a strategy that would assign the responsibility for implementing microfiber collection technology to municipalities and public services at sewage or water treatment plants before they discharge water into the aquatic environment. While this approach appears promising and ambitious, it also faces significant challenges due to the scale of these facilities. The existing technology is inadequate for managing such large volumes of water. Additionally, most available technologies primarily rely on filtration and separation methods, and there is currently no technology that effectively targets microplastics specifically.

Environmental and Health Impacts: Various policymakers and environmental protection organizations have expressed growing concern about microplastic pollution. Some reports indicate that acute health hazards are associated with microplastic exposure, necessitating comprehensive investigation. While chronic exposure to microplastics exhibits some degree of organ-specific toxicity, further research is essential before issuing any public advisories. In contemporary society, humans have the ability to limit their exposure to microplastics; therefore, evidence-based research is needed to determine the varying threshold levels of microplastics in different parts of Europe and around the world. This variation necessitates additional studies on chronic exposure toxicities. In contrast, aquatic life faces more severe consequences, as their ecological environments are significantly impacted. Organisms such as plankton, fish, and larger marine animals have been found to ingest microplastics, which may differ from the ingestion of microplastics from soil, as it is not in a dissolved form (as shown in Figure 6). Additionally, airborne microplastic pollution poses serious risks, particularly to lung health. Individuals working in environments where they are continuously exposed to microfibers should follow more rigorous occupational health protocols to mitigate these risks. Interestingly, microplastics have also been discovered in commonly used products such as food, beverages, and seafood, indicating their pervasive presence in our ecosystem. The long-term effects of microplastics remain largely unknown, as replicating similar conditions in laboratory animals is challenging compared to their natural habitats contaminated with microfibers. Moreover, in addition to the potential toxicity of microplastics themselves, some microfibers may contain dyes, by-products, or other chemicals—including catalysts and monomers—that can leach into the environment over time through various degradation processes, weathering, or unknown mechanisms, further illustrating the severity of microplastic toxicity. Most studies suggest that microplastics can damage the digestive and respiratory systems and may also provoke inflammatory responses. Elevated levels of microplastics may serve as carriers or support surfaces for microbes, potentially disrupting microbial ecology.

### 3.2. Water Pollution and Its Impact on Agriculture

One of the prerequisite requirements for establishing a textile industry is its location near a water source. Consequently, most dyeing industries are situated near rivers and lakes, particularly in countries with abundant water bodies, such as those in Southeast Asia. However, the processes involved in coloration or dyeing are complex and have a significant ecological footprint. These processes utilize various water-soluble chemicals, including alkalis, acids, salts, and dyes, resulting in wastewater that poses severe environmental hazards. Water is predominantly required in textile industries for several operations, including dyeing, printing, washing, bleaching, and mercerization. As a result, textiles rank second among industries in terms of water pollution, primarily due to their high-water consumption—approximately 230 to 270 tons of water is needed to dye one ton of textile material [105]. This clearly highlights the industry’s dependence on water. The textile industry, particularly those involved in dyeing cellulose-based fibers, is one of the largest producers of wastewater. Cellulose fibers typically require a significant amount of chemicals for preactivation and dyeing at an industrial scale. Typically, large quantities of reactive dye (which does not react with cellulose fibers) and other chemical auxiliaries (added to improve dye exhaustion or fixation) are discharged in the textile effluent (as shown in Figure 7). Cellulosic fibers make up about 30–32% of overall textile fiber usage or production (cotton~25%, regenerated cellulose-based fibers such as viscose, lyocell, and modal~6–7%). The solubility of dyes and fibers plays a crucial role in determining dyeing efficiency. For instance, anionic or cationic synthetic dyes (such as reactive, acidic, basic, and direct dyes) are inherently designed to dissolve easily in water. In contrast, the chemical characteristics of typical fibers can hinder the efficient dyeing of fiber (as discussed in Section 2.2 above). Dispersed synthetic dyes (mainly used for synthetic fibers) or vat dyes typically have limited aqueous solubility, which reduces their fiber fixation rate. While the addition of electrolytes can enhance dyeing, it also leads to an increase in chemical waste in wastewater. It is estimated that approximately 5% to 35% of dyes can be lost in wastewater. Although this loss can be mitigated by using specific chemical auxiliaries, many third-party textile manufacturers prioritize color shades based on samples provided by brands, often neglecting the issue of chemical waste. Consequently, a significant amount of chemical waste—approximately 200,000 tons annually—is produced by the textile industry. This chemical waste negatively impacts aquatic life, as colored wastewater reduces sunlight penetration, severely affecting marine ecosystems. Textile industries employ various remediation methods to mitigate the color components in their effluent, but none have demonstrated high efficiency. Most of these methods function as adsorbents or filters, and their effectiveness diminishes over time.

The textile mill produces a large volume of effluent containing various types of chemicals, primarily colorants, along with suspended particular matter that deteriorates the aesthetic quality of water and gives an unpleasant appearance with a foul odor. Most of these colorants are of organic aromatic chemotype and, therefore, produce significant long-term environmental damage. Environmental contamination occurs through leaching, runoff, and direct wastewater discharge. Once released into the environment, these dyes or chemical auxiliaries used in the dyeing process mix with water bodies. Wastewater contaminated with dye and related chemicals from textile industries reaches water streams, those that are used for potable drinking water can cross-contaminate and no method eliminates these pollutants which ultimately degrades the quality of drinking water, making it unsafe for consumption by human societies. Interestingly, the World Health Organization offers recommendations that only provide the overall safety and quality of drinking water but do not indicate tolerable thresholds for specific dyes present in the water [106]. In today’s world, more attention is provided to water quality and, therefore, many nations are coming forward to either ban or strictly regulate the specification of azo dye usage on the industrial scale. Notably, contamination with dyeing chemical waste is reported to reduce the soil quality and affect terrestrial plants, marine life, and ultimately, human societies. The draining of effluents from water bodies into farming lands can clog soil pores and alter the pH, which affects the solubilized form of nutrients (or plant supplements). This process eventually changes the soil texture, impacting soil quality and fertility. Additionally, the chemicals in these effluents can disrupt the osmotic balance, a key parameter for seed growth and germination. In some cases, this disruption reduces the availability of critical nutrients needed for plants to maintain an appropriate level of chlorophyll, thereby hindering their growth [107]. In my opinion, the chemical waste from these effluents can have either high or low pH, depending on the dyeing process used. This variation in pH can decrease the availability of nutrients, particularly the solubilized form of magnesium in the soil. Consequently, this could be one of the reasons why plants in these reported studies are found to have lower chlorophyll content. In a lab-scale phytotoxicity study, Ellafi et al. observed an adverse effect of Congo Red on the germination indexes and rates of seeds of *Raphanus sativus* (radish), *Lycopersicon esculentum* (tomato), and *Lepidium sativum* (cress) [108]. In my opinion, it is crucial to understand the concentration of pure dye in the samples, as the levels of dissolved dye in the environment can differ significantly. Additionally, the state of the dye can further change depending on variations in the pH. Therefore, it would be advisable for our research community to, if possible, use industrial samples. Another important aspect to consider is that the dye is not isolated but is discharged along with other chemicals. This interaction can alter the outcomes of lab-scale studies compared to real-world scenarios. Congo Red is banned in the textile industry due to its carcinogenic nature, making it unlikely to be found in significant concentrations in water bodies. Comparing the environmental toxicity profiles, terrestrial plants suffer from the drainage of chemicals in effluents. Meanwhile, aquatic plants are severely affected by the high presence of these chemicals in water channels. Moreover, surface saturation over water bodies blocks light penetration, which is essential for photosynthesis in aquatic plants. This blockage leads to anoxic conditions that can be lethal for aquatic life.

### 3.3. Aquatic and Terrestrial Toxicity of Dyed Wastewater

The persistent environmental contamination caused by the textile industry has been documented in various reported literature [26,109,110,111,112]. Some reports indicate that certain dyes accumulate in aquatic life forms at the cellular level [113,114,115,116,117,118,119,120], while high molecular weight chemicals derived from dyeing processes accumulate in sediments on the ocean floor, slowly leaching and solubilizing into the water. This suggests that sustained exposure can have significant effects on marine life and ecology. The exact mechanism is still unknown, but it seems that cellular dye uptake alters cellular control [117,121,122,123,124,125]. In some cases, this loss of cellular control appears to promote undifferentiated cellular division [117,126,127]. However, further studies are needed to confirm whether this undifferentiated cellular division, linked to the cellular update of textile dye, can lead to human cancers. Despite this, it is clear that the hydrolyzable forms or by-products derived from synthetic dyes do possess carcinogenic potential to some extent. These health concerns underscore the need for occupational risk assessments in the textile industry. Another argument relates to the concentration of dissolved forms and the consistency of exposure. In laboratory settings, cell cultures are exposed to doses continuously for durations that may differ significantly from real-world scenarios. As a result, predicting or extrapolating the metabolism or carcinogenic potential of chemicals from the textile industry and their by-products can be quite challenging.

Dyes are known to produce various types of cellular toxicity; however, more studies are required to fully understand their mechanisms. For example, reactive dyes that contain triazine or other reactive groups and are known to be used in cellulose fiber reactive dyeing, are associated with occupational allergy risks. Due to the presence of reactive groups in the structures of these reactive dyes, they are found to induce more toxicity compared to other dyes that do not have these reactive groups. Animal models have shown that these reactive dyes can be genotoxic, mutagenic, and carcinogenic. In comparison, dyes without reactive groups are likely to be less toxic because they lack these harmful components. However, further investigation is needed to confirm these findings. Azo dyes, in particular, can withstand a narrow range of pH where their structure remains unaffected by changes in pH. However, a drastic change in pH levels, whether higher or lower, can lead to their hydrolysis, releasing aromatic amines that may cause various types of organ or cellular toxicity. In conclusion, various reports support that azo dyes tend to hydrolyze in the acidic pH (change in pH), releasing aromatic amines that have toxicity. These aromatic amines are known to have the potential to induce different types of organ or cellular toxicity.

One might wonder why we do not replace the diazo chromophores in these commercial dyes with other chromophores that are less prone to hydrolysis and do not produce aromatic amines. For example, anthraquinones possess structural stability across a broader range of pH levels and can tolerate higher temperatures. However, the synthesis of these azo dyes has been refined over time and relies on readily available starting materials, while using other chromophores (such as anthraquinones) often entails a more complex multi-step synthesis that may result in lower overall yields. Although alternative chromophores offer advantages in terms of structural stability over chromophore-based diazo groups, there is currently a lack of data regarding their cellular toxicities. While other chromophores as replacements for azo-based chromophores do have advantages in terms of structural stability, there is no supportive information about their cellular toxicities. Thus, without comprehensive investigation, it is challenging to compare one chromophore to another or to assess the toxicity of dyes with different chromophore types. Notably, some naturally occurring metabolites have a diazo group. One of the key examples is the DNA intercalation nature of diazo groups of lomaiviticin and kinamycins. It seems that by the reduction of diazo groups, there is a loss of nitrogen leading to radical production (DNA-cleaving fluorenyl). Interestingly, one of the biochemical pathways involved in diazo-based functionality synthesis, L-aspartate-nitro-succinate, is actively expressed in various *Streptomyces* species [128]. Although homologous genes for this pathway are actively expressed in *actinobacteria*, understanding of their substrate chemistry still lacks enough supportive information. Importantly, it seems that the critical requirement of facilitating diazo formation depends on the in situ nitrite formation that is found to be formed from a nitrosuccinic acid with the help of redox enzymes [128].

Aniline derivatives and their impact on environment and human societies: Reductive hydrolysis of the diazo groups of the commercial azo dyes, results in the by-products (aromatic amines). These aromatic amines are known as carcinogenic chemicals. Aromatic amines are known commonly as anilines. Commercial processing (either manufacturing or their applications) of anilines is not only limited to the dye manufacturer or dyeing industries but also can be found in other industries (such as agrochemicals, rubber processing, and pharmaceuticals). Starting materials (chemicals) that are used for synthetic diazo-based dyes or colorants, are found in commercial applications in food, leather, and textile industries. Some of the common precursors used for the synthesis of commercial diazo-based dyes are acetanilide, phenylenediamines, and alkyl-substituted anilines.

Extensive use of anilines in industrial processing results in a high percentage of their release in wastewater. Based on toxicity studies of aniline, it was observed it has high interference with the oxygen-carrying protein of blood (hemoglobin). Anilines potentially convert hemoglobin into methemoglobin, significantly impairing its ability to bind oxygen and, therefore, reducing its oxygen-carrying capacity. This condition, known as methemoglobinemia, leads to cyanosis and hypoxia. The severity of methemoglobinemia depends on the extent of exposure. Additionally, the diazo group is highly reactive and can be easily activated (reductive hydrolysis) by enzymes. The resulting metabolites can contribute to cellular damage. For example, an enzymatic transformation of aniline to p-aminophenol and other reactive species can cause nephrotoxicity and hepatotoxicity. Continuous exposure to anilines leads to chronic toxicities. For example, occupational health risks are associated with the workers working in the dye and rubber industries as they have continuous exposure to benzidine and 2-naphthylamine which are reported as carcinogens. Reports indicate in several instances, these chemicals may cause bladder cancer, hematological disorders, and liver and kidney malfunctioning. There is also a high ecological risk with aromatic amines (anilines) as they can be found in soil, water, and air, affecting a whole ecosystem. However, their solubility in water depends on the presence of polar groups in their structure (such as phenolic and carboxylic groups), helping them to solubilize in aquatic environments. Anilines in their solubilized aqueous form in aquatic environments have shown high toxicity to marine life (ranging from fishes, algae, and invertebrates), where over a period of time, they accumulate in those living organisms and can enter the food chains. Consumption of contaminated seafood poses a severe threat to modern human societies. Additionally, anilines undergo photodegradation over time in the presence of light, which can lead to the formation of nitrosamine derivatives, known as carcinogenic chemicals.

In a study conducted by Khan et al. assessing the harmful effects of textile wastewater for irrigation purposes, they evaluated the availability of nutrients and their levels in the soil and the production rate of crops correlating with the impact of persistent azo dye contamination in the soil [129]. They used experimental tests, for example, the *Allium cepa* chromosomal aberration assay, the *Escherichia coli* DNA repair defective mutation assay, and the *Salmonella*/mammalian microsome test to evaluate the toxicity of soil irrigated with textile wastewater from the Panki industrial site (Kanpur, India). The Ames test and DNA repair defective mutation test results indicated that extracts of soil samples containing the azo dye caused varying degrees of DNA damage, suggesting potential involvement of mutagenicity and genotoxicity. Additionally, soil samples containing azo dye affected the mitotic index and caused chromosomal abnormalities. Importantly, the author isolated a bacterium (identified as *Ochrobactrum intermedium* by 16S rRNA gene sequencing) capable of tolerating reactive black 5 (500 μg/mL) in high concentrations and concentration of salt (20 g/L) while achieving 93% dye decolorization at 37 °C and in a pH range (5–9). GCMS analysis showed sodium-3-aminonaphthalene-2-sulfonate and sodium-2-hydrosulfonylethyl sulfate as degraded products. However, in my opinion, as GCMS analysis would be better suited for those samples containing low boiling points or lower molecular weight compounds or metabolites, it seems that LCMS could be a better option to analyze the degraded organic substructure as reactive black is known to produce high molecular weight components. This study highlighted the potential risks of irrigating with textile wastewater, emphasizing the need for toxicity assessment, and screening of different plant species that can sustain such a chemically toxic environment to achieve an effective phytoremediation.

In another study, Garcia et al. investigated the ecotoxicological effects of textile chemical waste contaminated with heterobifunctional cellulose fiber reactive dye (Reactive Red 239 dye, which possesses monochlorotriazine and vinyl sulphone group, therefore, exemplifying it as an example of heterobifunctional) on various aquatic life forms at different life stages, including adults and embryos of bacteria (*Vibrio fischeri*), crustaceans (*Daphnia similis*), and snails (*Biomphalaria glabrata*) [130]. They also assessed mutagenicity using the *Salmonella*/microsome assay (TA98, TA100, and YG1041 strains). By comparison, *Vibrio fischeri* bacteria were found to be very susceptible to Reactive Red 239 dye (EC_50_ = 10.14 mg/L), followed by mollusk embryos at all stages (EC_50_ = 116.41 to 124.14 mg/L), and *Daphnia similis* (EC_50_ = 389.42 mg/L), while least affected were adult snails (LC_50_ = 517.19 mg/L). All the tested organisms [E(L)C_50_ < 15%], with *Biomphalaria glabrata* embryos showing different responses at early stages (blastulae and gastrulae, EC_50_ = 7.60 and 7.08%) compared to advanced life stages (trochophore and veliger, EC50 = 21.56 and 29.32%), demonstrated toxicity towards to textile effluent. Likewise, developmental and sub-lethal impacts of chemical waste containing textile effluents were recorded in *Daphnia similis* and *Biomphalaria glabrata* embryos. No mutagenic observation was recorded during these studies. These findings highlight the significance of assessing the harmful effects of textile effluents on aquatic life forms across different trophic levels, highlighting the necessity of environmental protection and underscoring the urgency of effective treatment methods. Textile dyeing is known to produce large volumes of water and generates significant amounts of colored effluents containing hazardous chemicals, which can substantially alter aquatic environments and negatively affect organisms at various trophic levels.

The physicochemical characteristics of fiber are generally considered limited thereby requiring additional chemical auxiliaries to enhance their dyeing. In particular, the inert nature of cellulose fibers necessitates the use of reactive dyes, which are highly water-soluble and additionally require salt or electrolyte to facilitate the migration of dye molecules towards the fiber. This process generates large volumes of textile effluent contaminated with hazardous chemicals. Textile industrial waste, even from other fibers than cellulose fibers, is characterized by high chemical oxygen demand (COD) and biological oxygen demand (BOD) values due to soluble and insoluble forms of chemical auxiliaries (organic and inorganic), and hydrolyzed dyes, posing severe threats to aquatic and terrestrial ecosystem and lifeforms. Despite their inert nature, cellulose fibers require reactive dyes, which offer a wide range of shades and reasonable fastness properties and are widely available commercially. Using reactive dyes for cellulosic materials results in effluents with high levels of chemical and biological oxygen demand and dissolved solids. The use of alkali in aqueous dyeing increases the pollutant discharge, including total dissolved salts (TDS) and total suspended solids (TSS), posing significant removal challenges. Therefore, such high chemical waste in textile effluent significantly raises the cost of its treatment or recycling. Due to considerable development in understanding dye–fiber chemistry and the emphasis of the regulatory bodies on sustainable development goals in the last decade, most textile industries are implementing their processing towards sustainable dyeing practices. However, dyeing specific fibers or textile materials still produces wastewater containing unreacted dyes (or unfixed dye), hydrolyzed forms of dyes, and other chemical auxiliaries (acid/basic electrolytes). Cellulose-based fibers, in particular, still inherit dye–fiber chemistry issues leading to high chemical waste. Recent advancements have highlighted the significant potential of using living organism-based decolorization processes (such as plants or microbes) and technologies in mitigating the ecological impact of synthetic dyes. The success of methods such as successive aerobic–anaerobic digestion of textile chemical waste containing wastewater support the promising application of bioremediation as an effective approach to achieving *Sustainable Development Goal 12* (SDG 12), which is “*Ensure sustainable consumption and production patterns*”, and *Sustainable Development Goal 6* (SDG 6), which is “*promotes clean water and sanitation*”.

## 4. Methods for the Degradation of Textile Dyes

Textile effluents are known for high levels of BOD and COD, salinity, and TDS which are characterized by higher levels of synthetic dyes and other chemical auxiliaries making treatment of such diverse chemical waste, complex and challenging. To address water pollution from textile effluents, it is important to devise strategies that promote effective, efficient, and economical methods and technological advancements. Most of the reported approaches for treating textile effluent are based on utilizing the physicochemical parameters of materials/components of textile effluents, while only a handful of them indicate the inclusion of biological methods. Physicochemical processes [109,131,132,133,134], such as filtration, precipitation, mineralization, electrochemical destruction, ion exchange, adsorption, irradiation, and coagulation, are conventional and often tend to lose efficiency if continuously used for a prolonged time. Although industries find these methods simple and ready, maintenance requirements and increased operational costs severely hamper their continuous use during processing. For example, the chemical treatment method solely based on oxidation-based decolorization does the oxidative cleavage of diazo functionality of azo synthetic dyes using oxidizing agents such as hydrogen peroxide (H_2_O_2_), Fenton’s reagent, and sodium hypochlorite (NaClO) [135]. Importantly, oxidizing methods indeed have exhibited reasonable remediation (as a decoloration) of textile effluent but, due to their high energy consumption and their running cost, they are less cost-effective when operating them continuously for long periods of time. The sludge formation due to the accumulation of oxidized side products is found to be toxic and, therefore, requires a secondary treatment.

### 4.1. Bioremediation Mechanism: Study-Supported Theories

Biological methods have gained significant attention from researchers due to their numerous advantages over traditional methods, such as their operational ease, being eco-friendly and economical, high efficiency, and minimal sludge production (that could be toxic). Bioremediation is a type of biological treatment that uses living organisms to remove pollutants (such as dyes). This decolorization process could use plants (known as phytoremediation) and microorganisms such as bacteria, microalgae, and microfungi (known as microbial bioremediation). Interestingly, bioremediation can utilize natural or genetically modified organisms, their dead biomass, or their secretions [136]. In bioremediation, the chemicals (derived from dyeing effluent) break down into simpler compounds (such as water and carbon dioxide), and if those chemicals are considered as dye or dye components then the process can be called dye mineralization [137,138,139,140].

Biological methods are more versatile than conventional ones and are applicable both in situ and ex situ. Importantly, repetitive culturing of microbes can adapt to hostile environments, developing resistant cells in the solutions. These resistant cells can be sustained in a hostile environment (which contains high levels of dye concentrations and chemicals used in dyeing), indicating their utilization of these chemicals for their cellular metabolism. Therefore, effective dye bioremediation requires microorganisms that can withstand high dye levels and transform them into chemicals that have none-to-lower ecological risk [141,142]. In general, the degradation or hydrolysis of dyes (in their solution state) resulted in colorless solutions (loss of color intensity). This decoloration of a dye solution can be measured using spectrophotometric instruments, which are low-cost and easily available lab methods. One of the questions that remains unanswered is about the mechanism by which these microbes facilitate dye degradation. Some preliminary investigations suggest that the dye decolorization may occur through a mixed pathway including cellular adsorption and metabolism. However, a comprehensive investigation is needed to fully understand these processes.

Dye uptake and role of liposaccharide in cell wall: One of the mechanisms of dye uptake by microbes can be explained with the biosorption process where contaminants that are present in textile wastewater effluents are absorbed by the cell wall and stored for a period of time. However, in my opinion, this accumulation within the cell wall is mostly dependent on pH and osmotic pressure differences. One of the factors that may play a vital role in storing these dyes within the cell wall is the lipopolysaccharide content percentage of the microbes. Therefore, the ionic groups of the dye molecules (for example, -SO_3_Na, -OH, -NH_2_, -COOH) assist them in stabilizing within the cell wall as they may exhibit ionic interactions with the polar groups available in lipopolysaccharide (which contain hydroxy, amino, phosphate and carboxy groups). However, to estimate the significance of osmotic pressure, and possibly the complexation between inorganic/organic components, one study supports such information. Sarim et al. reported a pseudo-second-order kinetics for *Bacillus subtilis* KK01 that can efficiently (92.8%) adsorb Congo Red dye at 30 ppm dye concentration [143]. However, pretreatments can enhance adsorption abilities; for example, those chemicals can modify cell surface properties, while autoclaving can increase surface area by rupturing the microbial cells. Dead cells are often known as better biosorbents than living cells because of their non-living nature they do not require a nutrition cycle and, therefore, do not alter the cellular metabolism and could accumulate dye for longer periods of time. Furthermore, factors like shaker speed, pH of the solution, fluctuations in temperature, the effective surface area of dye in contact with the cell wall of the microbe, exposure time, and dye contamination levels significantly affected the biosorption process [144].

As the dye molecules saturate the cell surface, preventing further adsorption [145], in my opinion, the dye absorption rate will decrease over time because dead cells lack cellular activity. As a result, these dead cells are unable to metabolize the accumulated dye molecules. Therefore, biosorption-based microbial models have scalability limitations, and therefore, their commercial success is questionable in their current state. Interestingly, some novel approaches are also implicated such as the inclusion of alternating current (AC)-driven bioelectrodes to decolorize/degrade the dyes in the solution. This bioelectrode optimizes electron transfer and, therefore, ultimately enhances synthetic diazo dye degradation. Yuan et al. (2024) devised a low-voltage, low-frequency alternating current bioelectrode that significantly improves yellowed-colored dye (alizarin yellow R) degradation and mineralization by fostering a microbial cocktail providing ample genes or enzymes that ultimately perform an efficient dye degradation [146]. Research investigating the high dependence on co-substrates (of glucose and yeast extract) while maintaining optimal pH and temperature, is critical for achieving an efficient decoloration [147,148,149,150,151]. Additionally, microbes, particularly bacteria, often require supplementary carbon supplies since dyes (synthetic or natural dyes) are hard to break down and mineralize to elementary levels, where their mineralized metabolites can be used as a precursor in the carbon nutrient cycle of bacteria.

Although the kinetics of decolorization studies depend on many parameters, they are generally described as first-order kinetics or pseudo-second-order kinetics. For example, in a 10-day study, *Bacillus rigidiprofoundi* cells showed an 80% adsorption capacity for Remazol Brilliant Blue when incubated at a temperature (40 °C) and in a slightly acidic environment (pH = 5.5). Moreover, Fourier transform infrared spectroscopy (FTIR) and scanning electron microscopy (SEM) confirmed dye decolorization, while phytotoxicity assays indicate the degraded side products exhibited no toxicity [152]. Similarly, *Yarrowia lipolytica* (a yeast that is known to utilize hydrocarbons for its carbon cycle) can be an interesting choice, exhibiting pseudo-second-order kinetics and the Langmuir isotherm adsorption of Remazol Brilliant Blue dye, with both unmodified and chemically modified biomass, showed similar capacities. Within the first 15 min, the absorption of Remazol Brilliant Blue dye reached nearly 50%, while equilibrium was achieved after 6 h in a highly acidic environment (pH = 2.0) [153].

In contrast, the microbial biodegradation of diazo dyes indeed involves biocatalysis, converting dye structures (native or hydrolyzed) into simpler, low-to-non-toxic chemicals that can be easily consumed or participate in the nutrient cycles to be anabolized into useful metabolites. Importantly, the most common pathway for the diazo dyes involves the break down from their diazo (-N=N-) functionality, either by a specific enzymes or microbial consortia, releasing simpler structures or molecules such as water and gases. For instance, azoreductases facilitate the degradation of diazo dye utilizing a redox mechanism catalyzing the reductive cleavage of the diazo bond, resulting in aromatic amine-based hydrocarbons. Some of these aromatic amine-based hydrocarbons exhibit aqueous solubility as their molecular size decreases, leading to a reduction in the structure bulkiness while retaining the sulfate groups, which further enhances water solubility.

### 4.2. Role of Bacteria in Bioremediation of Dye Solutions: Initial Challenges or Limitations

Exploration of bacterial species for dye decolorization began in the 1970s with the discovery of strains of *Bacillus* sp., and *Aeromonas* sp., and many more have been identified in recent years as those that perform an effective decolorization of diazo dye-based solutions/samples or have preferential selectivity for a specific dye chemotype. In comparison, bacterial bioremediation is considered more suitable than other microbial bioremediation methods, such as those involving microfungi and microalgae. This preference is due to the availability of diverse bacterial species, the ease with which their genetic modification or specific gene expression can be achieved, as well as their rapid growth and simpler cultivation methods. Additionally, some bacteria are known to grow even in extreme conditions, tolerating high pH or temperature, and salinity levels. In terms of implication for the bioremediation of a broad class of diazo dyes, bacteria often exhibit high decolorization efficiency. For example, under optimal conditions, *Enterobacter* sp. strain demonstrated a highly efficient decolorization of the solution containing three diazo dyes (Acid Red, Congo Red, Methyl Orange) with reductions in levels of TDS, COD, as well as concentration of sulfate, and salt levels while degraded by-products exhibit lesser phytotoxicity [154].

Various bacteria strains have been investigated to achieve efficient dye biodegradation, for example, *acinetobacter*, *klebsiella*, *bacillus*, *enterococcus*, and *staphylococcus*. It is quite evident that these bacterial strains certainly break the diazo bond of the synthetic dye using azoreductase enzymes or similar functional enzymes. In static incubation, at a temperature of 37 °C, *Staphylococcus* sp. efficiently degraded Remazol Brilliant Blue dye (100 ppm), with peroxidase and laccase enzymes playing key roles in the process. UV–Vis and mass spectrometry confirmed microbial degradation [155]. Additionally, another study using UV–Vis spectroscopy and FTIR analysis suggested a full degradation of methyl orange dye (150 ppm) at a temperature of 35 °C and neutral pH (7.0) within 12 h using *Bacillus* sp, while the degraded solution exhibiting no toxicity [156]. Singh and Dwivedi (2020) found that *Aspergillus terreus* GS28 facilitates the decolorization with an efficiency of 98.4% of Direct Blue-1 dye at 30 °C after 168 h of incubation, with manganese peroxidase and laccase enzymes playing a role, while degraded by-products were analyzed using GCMS profiles [157]. Srinivasan and Sadasivam reported autochthonous strains for three diazo dye degradation (Remazol Yellow RR, Joyfix Red RB, and Reactive Yellow F3R) using adapted and non-adapted bacteria strains of *Aeromonas hydrophila*. In their study, they found slower decolorization rates for non-adapted *Aeromonas hydrophila* MTCC 1739 compared to the textile-effluent adapted *Aeromonas hydrophila* SK16 bacteria. They utilized HPLC and GCMS analysis to study and evaluate the degraded by-products [158].

### 4.3. Anaerobic Bioremediation of Dye Solutions Using Microbial Bacteria

The reductive cleavage during anaerobic digestion of the diazo bond (-N=N-) leads to decolorization of the diazo dye solution and occurs through electron transfer. Synthetic diazo dyes contain various functional groups in their structures, which may be electron-deficient or electron-withdrawing groups (such as monochlorotriazine ring, halogen, carboxylic, and nitro groups). Therefore, redox substrates or their reducing forms, for example, flavin mononucleotide, flavin adenine dinucleotide, and nicotinamide adenine dinucleotide phosphate donate electrons to these groups, leading to the breakdown of the diazo bond during anaerobic digestion [159].

In 2018, Li and colleagues reported the anaerobic degradation of cationic Red X-GRL dye by *Shewanella oneidensis* MR-1 [160]. This electrochemically active microbe showed high decolorization, removing 100 ppm of the dye after 12 h of incubation. Citric acid and lactate served as electron donors in this anaerobic process. Phytotoxicity assays indicated that the decolorized dye-degraded products were non-toxic, with negative genotoxicity [160]. To enhance the anaerobic decolorization process, new methodologies were investigated. For example, in 2022, Nguyen et al. added Iron(III) oxide to improve microbial decolorization under anaerobic conditions. Their study, conducted in an anaerobic baffled reactor with 10 g/L Iron(III) oxide, 18.6 h of retention time, and at 30 °C for 170 days, showed significant enhancement in the decolorization process [161].

### 4.4. Aerobic Bioremediation of Dye Solutions Using Microbial Bacteria

The decolorization of diazo dyes is facilitated by oxygenase-like enzymes in aerobic conditions. The most probable ones are monooxygenase and dioxygenase. To commence the enzymatic hydrolysis of the diazo bond requires the transfer of oxygen atoms into the aromatic rings of dye molecules [162]. However, in my opinion, it seems that this decolorization or bioremediation could result from oxidative cleavage of the diazo bond of these dyes, therefore, there could be the possibility of involvement of oxygenase (or specifically, those enzymes from the peroxidase class). However, such information requires additional supporting details and a more thorough investigation. Notably, the presence of oxidized functional groups or those that contain higher oxygen atoms (for example, sulfonic and nitro groups) may affect the kinetic rate of aerobic-based decolorization of diazo dyes.

A complete decolorization of Remazol Navy Blue was reported using microbial fuel cells, specifically in an anodic chamber, suggesting the involvement of aerobic conditions [163]. The bacterial consortium followed first-order kinetics, achieving a complete decolorization of dye concentrations ranging from 25 to 100 ppm within 12 h of incubation.

Based on analytical data, the authors suggested that reductive cleavage played a role in the hydrolysis of diazo bonds, occurring during sequential aerobic stages of dye degradation. This finding was somewhat unexpected, as hydrolysis-based oxidative cleavage would typically be anticipated in such conditions, whereas reductive cleavage requires a source of reducing equivalents and is more commonly associated with anaerobic processes. However, other mechanisms may be involved in this process, rather than just one; therefore, further studies are needed to fully understand what exactly happens in the process, while the author observed no toxicity during their phytotoxic studies, indicating that dye mineralization or degradation indeed took place [163].

Seyedi et al. studied the aerobic digestion of Reactive Red 152 and Reactive Black 5 using haloalkalophilic bacteria [164]. This is quite an interesting study as both (Reactive Red 152 and reactive black 5) are homobifunctional reactive dyes. Reactive Red 152 has two monochlorotriazine cores compared to the two vinyl sulphonyl functionality of reactive black 5 as reactive groups. The aerobic incubation which required nitrogen and carbon enrichments, maximum decolorization was achieved for 50 ppm concentration at neutral pH, while a significant decolorization was exhibited with microbial consortia which consisted of three bacteria [164], while Fareed et al. reported that aerobic bacteria facilitated the decolorization of two diazo dyes (Reactive Black 5 and Reactive Orange 16) [165]. Importantly, the immobilized form of bacteria was found to achieve higher decolorization than their free cellular forms, while degraded products produced during the decolorization process were found less phytotoxic and bacteriotoxicity compared to the tested samples which contained native diazo dyes [165].

### 4.5. Anaerobic/Aerobic Bioremediation of Dye Solutions Using Microbial Bacteria

Sequential decolorization using both aerobic and anaerobic processes is generally considered a more effective choice and strategy than using either process alone. Research shows that the by-products of microbial decolorization can produce substituted aromatic amines, while a number of these by-products and specific by-product formations depend upon the parent dye structure. These aromatic amines are known to have various cellular and organ-specific toxicities. Therefore, a highly efficient bioremediation procedure is necessary to remove all dye or dye-like components from textile effluents. Although sequential anaerobic–aerobic bioremediation can achieve mineralization of the degraded by-products from the hydrolysis of diazo dyes, current applications of microbial bioremediation primarily focus on achieving the maximum efficiency for the dyes themselves, rather than their hydrolyzed by-products. As an example, plant–microbe action for the sequential anaerobic–aerobic bioremediation of diazo dyes can be used as a model. When tested, this method achieved a decolorization efficiency of approximately 53% under anaerobic conditions and 92% under aerobic conditions with a net removal efficiency of 98.79% for intermediates. Furthermore, the aromatic amines that formed during the degradation of diazo dyes are transformed, reducing their overall toxicity (supported by the phototoxicity assessment) [166]. A full elementary decomposition or degradation of diazo dyes certainly removes the toxicological profiles of first-stage diazo dye hydrolysis either by aerobic or anaerobic bioremediation. Intermediates possessing toxicological profiles such as substituted aromatic amines, could pose ecological risks if they remain unattended.

Numerous reports have prioritized their research to perform the bioremediation on these intermediates (or by-products) that result from the hydrolysis of diazo dyes to improve sustainability. A sequential anaerobic–aerobic digestion using eighteen different bacterial strains in a bioreactor achieved 91% decolorization for Acid Red 14. Importantly, the HPLC profiles supported that anaerobic decolorization involved azo bond cleavage and aromatic amine formation. Subsequent aerobic digestion exhibited 63% removal of amines that were formed as intermediates, such as 4-amino-naphthalene-1-sulfonic acid, indicating efficient decolorization and bioremediation on by-product intermediates formed after the decoloration of diazo dyes [167]. Interestingly, under anaerobic–aerobic digestion, a thermo-alkalophilic microbial consortium had shown sequential anoxic/oxic of diazo dyes, reaching 98% decolorization efficiency under optimal conditions [168]. Such studies pave the way for new strategies to achieve sequential bioremediation, reducing chemical waste and improving sustainability. However, the application of biotechnological advancements, such as using microbial consortia in bioreactors, can help researchers fine-tune processes to preferentially select one pathway over another, leading to more efficient bioremediation.

Several enzymes have been recognized for their crucial role in the bioremediation of textile dyes. A prominent example is the laccase enzyme, which is known to degrade various commercial textile dyes, including azo dyes from other industries [169,170,171,172,173,174,175]. However, there are exceptions where dye structures lack diazo functionality yet can still be hydrolyzed by enzymes. One such example is Indigo Carmine, a water-soluble blue acid dye. While, Indigo Carmine cannot be considered as a true azo dye, but it shares a similar ionic character as commonly associated with azo dyes, due to the presence of two conjugated indole rings in its structure. A study conducted by Bento et al. from the Department of Chemistry at the University of Aveiro (Aveiro, Portugal) explored the use of ionic liquids based on surfactant chemotypes to degrade indigo carmine [176]. In their experimental setup, they used laccase (~1000 U/L) alongside ionic liquids at varying concentrations (10, 50, 100, 250, and 350 mM) [176]. The ionic liquids utilized three types of surfactants: tetraalkylammonium-based cationic surfactants, imidazolium-based cationic surfactants and cholinium-based anionic surfactants. A control group without ionic liquids (pH 4.5) was also included in the study.

The findings revealed that laccase-based degradation was significantly influenced by the type of ionic liquid used, particularly in relation to its chemotype (whether it was anionic or cationic) and concentration. Ionic liquids with shorter alkyl side chains allowed for approximately 85% laccase activity. This observation suggests that the laccase enzyme’s performance may depend on hydrophobic interactions that affect its conformational structure, thereby influencing its catalytic activity [177,178]. Another important factor could be micelle concentration, which also impacts decolorization activity. Conversely, laccase activity was notably lower in the presence of anionic surfactants, achieving only about 50% decolorization compared to the control. This could indicate that cationic amino acids, such as arginine and lysine, play a crucial role in facilitating enzyme catalysis. Additionally, the anionic nature of indigo carmine likely faced greater repulsion when interacting with anionic surfactants, which may explain the observed underperformance.

Previous research has indicated that the initial binding between enzymes and anionic surfactants occurs at the positively charged amino acid side chains, promoting enzyme unfolding and the formation of enzyme–anionic surfactant complexes [179]. This process, in turn, reduces the availability of free enzyme concentration for catalysis. Liu et al. also questioned the proportional relationship between the polarity of ionic liquids and laccase activity [180]. Furthermore, the stability of the enzyme in such solutions remains a topic of debate, as “chaotropic cations and kosmotropic anions stabilize the enzyme, whereas kosmotropic cations and chaotropic anions tend to destabilize it [180]”.

Iark et al. studied the enzyme-based degradation of Congo Red, which is an azo dye and also an acid dye. The authors isolated a 41 kDa laccase enzyme from the white-rot fungus *Oudemansiella canarii* using a substrate mixture of sugarcane bagasse and wheat bran [181]. They achieved a decolorization efficiency of 80% with a concentration of 50 mg/L Congo Red within 24 h at a temperature of 30 °C and a pH of 5.5. The enzyme kinetics exhibited Michaelis–Menten parameters, with Km and Vmax values of 46.180 ± 6.245 µM and 1.840 ± 0.101 µmol/min, respectively [181].

The class of reactive dyes is often linked to high environmental toxicity, and Drimaren Red CL-5B is one such dye. A study demonstrated that non-adapted bacteria, specifically *Aeromonas hydrophila* MTCC 1739 and *Lysinibacillus sphaericus* MTCC 9523, were able to degrade Drimaren Red CL-5B at a concentration of 100 mg/L in cultivation media at 37 °C and pH 8.0 [182]. This degradation occurred under sequential aerobic–microaerophilic conditions over a period of 72 h. Molecular modeling showed that the binding energies of laccase and azoreductase from *A. hydrophila* and *L. sphaericus* with Drimaren Red CL-5B: −31.9921 kJ/mol and −18.1289 kJ/mol for *A. hydrophila*, and −27.2792 kJ/mol and −2.5185 kJ/mol for *L. sphaericus* as shown in Figure 8 and Figure 9. Importantly, *A. hydrophila* MTCC 1739 achieved 91.96% decolorization, while *L. sphaericus* MTCC 9523 reached 88.35% [182].

Laccase complex with Drimaren Red CL-5B: The structure of Drimaren Red CL-5B includes two naphthalene aromatic rings conjugated with a diazo bond. These naphthalene moieties are composed of sulfate groups, while the terminus of Drimaren Red CL-5B contains a vinyl sulphone. The sulfate group of the vinyl sulphone terminus of Drimaren Red CL-5B exhibits hydrogen bond acceptor interactions with Glu162 and the guanidine moiety of Arg229 of laccase enzyme. Meanwhile, the substituted sulfate groups on the naphthalene core have hydrogen bond acceptor interactions with the guanidine of Arg236 and the imidazole ring of histidine (His38 and His70) [177,178,179,180,181].

Azoreductase complex with Drimaren Red CL-5B: The sulfate groups of the naphthalene moieties of Drimaren Red CL-5B form hydrogen bond acceptor interactions with the -NH- of Asn97, the phenolic group of Tyr96, the alcoholic -OH side chain of Thr18, the guanidine of Arg17, and the backbone of Gln20. However, no interaction of the vinyl sulphone terminus of Drimaren Red CL-5B was observed with any amino acid in the catalytic site of the Azoreductase enzyme.

Laccase (*L. sphaericus* MTCC 9523) with Drimaren Red CL-5B: This complex showed that the sulfate groups on the naphthalene rings of Drimaren Red CL-5B have hydrogen bond acceptor interactions with Cys34 and Ala242, while the polar phenolic group on the naphthalene showed a hydrogen bond donor interaction with His32. The vinyl sulphonyl terminus of Drimaren Red CL-5B exhibited a hydrogen bond donor interaction with the thiol side chain of Cys110. Additionally, Arg244 in the catalytic site interacted with the triazine core of Drimaren Red CL-5B through a hydrogen bond donor interaction via its guanidine side chain.

Azoreductase (*L. sphaericus* MTCC 9523) complex with Drimaren Red CL-5B. Although no hydrogen bond acceptor interaction was found with the sulfate groups of the naphthalene core of Drimaren Red CL-5B, the phenolic OH was able to show a hydrogen bond donor interaction with Lys18. Interestingly, the triazine core of Drimaren Red CL-5B was able to show a hydrogen bond acceptor interaction with the guanidine moiety of Arg11 and the indole -NH- of Trp93, while no interaction of the vinyl sulphonyl terminus of Drimaren Red CL-5B was observed.

## 5. Biodegradation of Fibers

Degrading textile fibers is challenging due to the diverse chemistries involved, making it difficult to develop a universal degradation strategy for all fiber materials. While chemical cocktail-assisted degradation has shown promising results, scaling up such procedures could pose more environmental hazards than benefits for recycling textile fibers. For instance, the hydrolysis of cellulose or protein fibers can be facilitated by enzymes, as these homopolymers consist of monomeric units linked by glycosidic linkages or amide backbones, which are relatively easy to hydrolyze. However, synthetic fibers pose more challenges because their monomeric units are covalently bonded, making it difficult to find enzymatic methods to break these bonds. Nevertheless, using a cocktail of enzymes can be beneficial, as seen with cellulose acetate, which involves a two-step degradation process: first by esterases, followed by cellulases.

### 5.1. Biodegradation of Natural Fibers

#### 5.1.1. Biodegradation of Cellulose

Saxena et al. from G B Pant University (India) in 1992 reported saccharification and fermentation of waste newspaper to produce ethanol [183]. They used a newspaper containing 85% cellulose, 12% lignin, 2–7% moisture, and 2–3% ash. They found that an alkali treatment of the paper increased the extent of saccharification, from 9.2 to 50.4%, using a cellulase enzyme for 3 days of incubation. Importantly, when cellulase (*Trichoderma reesei* QM 9414) was tested on alkali-treated newspaper, the saccharification reached 55.8% after 3 days of incubation. The authors’ findings on saccharification level were in agreement with Andren et al. who also recorded a 50% transformation of wastepaper using cellulase originating from *Trichoderma reesei* [184]. An interesting observation was recorded where steadily increasing substrate proportionally a corresponding increase in the production of ethanol was noted. After 4 days of fermentation, the ethanol remained constant until 5 days, and exhibited no change in the last 24 h, indicating either nutrient exhaustion or negative cooperativity of produced ethanol. To further support the observation of negative cooperativity of produced ethanol, a study performed by Ghose et al. revealed cellulase (*Trichoderma reesei* QM 9414) enzymatic activity noncompetitively inhibited by ethanol [185], while Ooshima et al. found no impact on the *β*-glucosidase and *endo*-glucanase when the concentration of ethanol is less than 4M [186].

Another study by Kuo et al. investigated the saccharification of cellulose-based materials. In their study, they used wet pretreated waste textiles for the enzymatic saccharification reaction. A total of 0.5 g dry-weight waste textiles was mixed with a phosphate buffer (pH 5, 100 mM) filtered on filter paper, which was subjected to treatment with Cellulase AP3 (10 FPU/g) of *Aspergillus niger* strain for hydrolysis. This process maintained a temperature of 50 °C, stirring at 250 rpm for 48 h, resulting in the production of 60–91% reducing sugars compared to the 25–35% for nontreated samples. The soluble reducing sugars release was continuously monitored, while the reducing sugar content of Cellulase AP3 was measured initially as it contains reducing sugars as well [187].

Quartinello et al. studied the grinding of samples to 1mm to enhance the surface area for enzymatic treatment and improve mass transfer [188]. Similarly, the textile samples were prepared (1g) and incubated with sodium citrate buffer (pH 4.8) possessing a cellulase cocktail (2750 U/mL) maintaining a temperature of 50° C with stirring at 400 rpm for 5 days, while the quantification of glucose was carried out by using Agilent 1260 Infinity II HPLC, with IC SEP-ION-300 coupled with refractive index detector with a flow rate of 0.325 mL/ min and temperature was set up at 45 °C [188].

#### 5.1.2. Biodegradation of Silk and Wool

Silk and wool are protein fibers and, therefore, belong to the natural fiber class. Mainly protease enzymes have shown a higher affinity for degrading silk and wool due to their peptide nature and producing α-amino acids. Different conditions and sources of protease have been found to facilitate the peptide hydrolysis. Li and colleagues investigated the in vitro enzymatic degradation of porous silk fibroin sheets using α-chymotrypsin, collagenase IA, and protease XIV [189]. When a 9 cm² freeze-dried silk fibroin sheet was incubated with protease XIV (EC 3.4.24.31; 1.0 U/mL) at 37 °C and pH 7, a 70% degradation was observed after 15 days. Treatment with collagenase IA (EC 3.4.24.3) resulted in a decrease in the Silk II crystalline structure, while a small amount of Silk I crystalline structure was recorded. Interestingly, the high Silk II form disappeared with protease XIV treatment, but overall crystallinity increased due to the rise in Silk I. The pore size of the fibroin sheets increased with degradation time and more than 50% of the degradation products were free amino acids from protease XIV treatment [189].

Navone et al. from Queensland University of Technology (Brisbane, Australia) studied selective biodegradation of wool fibers from wool/polyester blended fabrics using an enzymatic approach with the determination of soluble peptides and percentage degradation [190]. In their study, they used protease onto a fabric sample (0.1 g of 5 mm × 5 mm) which was fractioned into a sample sized ≤0.6 mm, subjected to treated with 2, 4 and 10 KU/mL (One keratin unit (KU) is defined as the amount of enzyme that causes an increase of 0.1 in absorbance at 595 nm after 1 h of incubation at 37 °C) of protease (pH 10 for 16 h at 50 °C at 200 rpm) where more than 95% of weight loss was measured [190].

### 5.2. Biodegradation of Synthetic Fibers

Recent studies suggest over 359 million tons of plastics are used globally per year. Nearly 150–200 million tons accumulate in the natural environment or landfill [191]. Poly(ethylene terephthalate), more commonly known as PET is the most used plastic, with 70 million tons used in textiles and packaging industries every year worldwide. However, thermomechanical recycling procedures are the primary methodology used, which results in a loss/reduction of the material’s mechanical properties [192]. However, the high presence of the aromatic nature of terephthalate units significantly decreases the chain mobility, making it relatively difficult to hydrolyze [193]. Despite these challenges, some research has been conducted on achieving the biodegradation of synthetic fibers, as mentioned in Table 2.

#### 5.2.1. Biodegradation of Cellulose Acetate

Haske-Cornelius et al. studied cellulose acetate biodegradation with the help of enzymes [194]. Since cellulose acetate is a typically acetylated form of cellulose, therefore the first step of their degradation is usually deacetylation, followed by hydrolysis into glucose units. The authors investigated deacetylation using HPLC and HPLC-MS-TOF, where testing esterases capable of deacetylating triacetin (glycerol triacetate) and glucose pentaacetate (which they used as a model compound for cellulose acetate) [194]. The most effective esterase for deacetylation belongs to enzyme family 2 (AXE55, AXE53, GAE), capable of deacetylating cellulose acetate with a degree of acetylation up to 1.8. Additionally, when esterase and cellulases are used together, they exhibit synergism, increasing absolute glucose recovery from cellulose acetate (1.8) from 15% to 28%. While chitinase showed no activity, Lytic polysaccharide monooxygenase (LPMO) and cellobiohydrolase could cleave cellulose acetates with a degree of acetylation up to 1.4. Importantly, the distribution of acetyl groups, degree of substitutions, and chain length, play a crucial role in influencing cellulose acetate degradation. This study suggests that a successful enzyme-based deacetylation system requires a cocktail of enzymes that can randomly cleave and generate shorter cellulose acetate fragments. In an experiment where the authors wanted to use esterase followed by cellulase, they found the deacetylation of cellulose acetate was best with AXE 55 esterase. In this experiment, a cellulose acetate with 1.4 D.S. (7.54 mg·mL^−1^ in 20 mM Tris-HCl buffer at 25 °C pH 8.5) for the esterase (34 µg·mL^−1^) treatment incubated for 165 h, and 80% deacetylation was achieved. While, 48% glucose recovery using Cellic^®^ CTec3 multicomponent cellulase was recorded in conditions (pH = 5.0, temperature = 55 °C, and incubation lasted 168 h) [194].

#### 5.2.2. Biodegradation of Polyethylene Terephthalate (PET)

Ren Wei et al. synthesized the nanoparticles (mean diameter 100–160 nm) from samples of polyethylene terephthalate with different crystallinities to enhance the exposure of such samples for enzymatic hydrolysis [195]. A decrease in turbidity correlated to the enzymatic degradation via surface erosion, enabling the author to directly assess the kinetic rates of polyethylene terephthalate enzymatic hydrolysis. The author used *Thermobifida fusca* KW3 cutinase (50 μg/mL) for 0.5 mg PET fiber nanoparticles, incubating them at 60 °C at a pH of 8.5, with stirring at 1000 rpm, while after 40 min, a more than 97% reduction in turbidity was recorded. However, differential scanning calorimetry detected an increase in polydispersity of enzyme-treated polyethylene terephthalate nanoparticles, highlighting the possibility of an endo-type hydrolytic mechanism [195]. Tournier et al. described a leaf–branch compost cutinase variant (3 milligrams per gram of PET) that achieves polyethylene terephthalate over 90% depolymerization forming ethylene glycol and terephthalic acid [196]. Ronkvist et al. from Polytechnic University, (New York, NY, USA) studied a PET film (15 × 15 mm^2^) with a thickness of 250 μm using different cutinases (*Pseudomonas mendocina*, *Humicola insolens*, and *Fusarium solani*) [197]. Additionally, they compared the catalytic activities of various cutinases using low crystallinity and biaxially oriented polyethylene terephthalate films as model substrates. *Humicola insolens* exhibited reasonable thermostability with maximum initial activity from 70 to 80 °C, while *Pseudomonas mendocina* and *Fusarium solani* found high catalysis at 50 °C. Importantly, the studies showed that cutinase was found to achieve 10-fold more activity for low crystallinity polyethylene terephthalate films than biaxially oriented polyethylene terephthalate film (which had 35% crystallinity). In conclusion, *Humicola insolens* catalyzed the low crystallinity polyethylene terephthalate films at 70 °C 97 ± 3% weight loss in a 96 h study.

**Table 2 polymers-17-00871-t002:** Examples of enzymes catalyzing the depolymerization of natural and synthetic fibers. The table was adapted from Egan et al.’s work [198].

Fiber/Polymer	Fiber-Type	Expected Depolymerization Products	Enzymes	Ref.
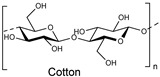	Natural fiber	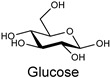	Cellulase AP3 (*Aspergillus*)	[187]
Cellulase: cellulase cocktail	[188]
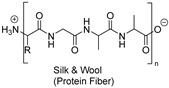	Natural fiber	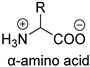	Protease: Protease XIV, 1 for silk fibroin sheetcollagenase IA	[189]
Serine Protease: Ronozyme^®^	[190]
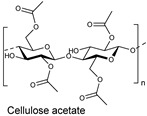	Synthetic fiber	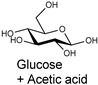	Esterase and cellulase: Synergism	[194]
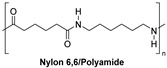	Synthetic fiber	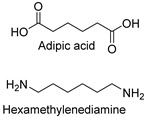	*Nylon hydrolase*: 6-aminohexanoate oligomer hydrolase	[199]
*Protease*: Protex^®^ modified subtilisin, Lipase: Lipex^®^ serine hydrolase	[200]
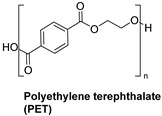	Synthetic fiber	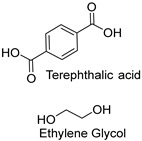	*Cutinase* (*Thermobifida fusca* KW3)	[195]
*Cutinase* (Leaf–branch compost cutinase variant)	[196]
*Cutinase* (*Humicola insolens*)	[197]
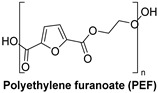	Synthetic fiber	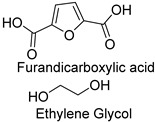	*Cutinase* (*Humicola insolens*)	[201]
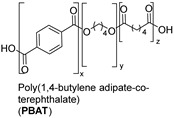	Synthetic fiber	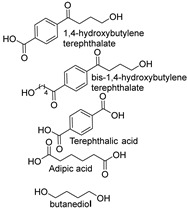	*Lipase*: Pelosinus fermentans	[202]
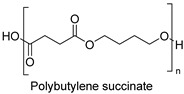	Synthetic fiber	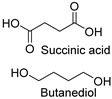	*Cutinase* (*Fusarium solani*)	[203]
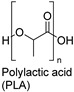	Synthetic fiber	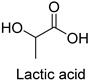	*Protease*: (*Tritirachium album*)	[204]
*Protease* (Proteinase K)	[205]
*Protease* (*Bacillus subtilis*)	[206]
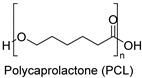	Synthetic fiber	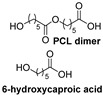	*Cutinase* (*Humicola insolens*)	[207]
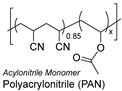	Synthetic fiber	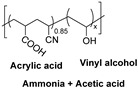	*Esterases*: (*Fusarium solani* pisi)	[208]
*Nitrile hydrolyzing enzymes* (*Micrococcus luteus* strain BST20)	[209]
*Nitrilase*: Commercial Cyanovacta Lyase	[210]

## 6. Conclusions

Fiber chemistry is a dynamic and interdisciplinary field that integrates elements of organic chemistry, polymer science, and material engineering. Understanding the molecular structure and properties of fibers enables scientists and engineers to develop new materials with tailored characteristics for a wide range of applications. From everyday clothing to advanced aerospace materials, fibers play a crucial role in modern life, and ongoing research is expanding their potential applications. However, the textile industry is known for producing a large amount of wastewater effluent (contaminated with dye materials). These dyes continuously leach out throughout their lifecycle, accompanied by the release of microfibers. To address these challenges, it is essential to understand the mechanisms involved in the leaching process and microfiber release. Different fibers used in the textile industry release varying amounts of microfibers, as explained in the sections on dye–fiber chemistry in this manuscript. Effluents containing dye materials present another pressing issue, as dyes are specifically designed for particular fibers, posing challenges that require solutions rooted in fiber chemistry. Recently, the application of enzymes and microbes for degrading fiber and dye effluents has gained attention. Since cellulose-based fibers constitute a significant proportion of the materials used in textile industries, therefore, new dyeing techniques and degradation technologies are needed during their industrial processing. The use of enzymatic or microbial cocktails shows promise in the bioremediation of such materials, but achieving efficiency and scaling at the commercial level remains a challenge. Therefore, green chemistry strategies and material substitution are essential for developing safer alternatives. For example, many synthetic dyes use diazo-based chromophores, which produce toxic aniline derivatives that can infiltrate the food chain, ultimately disrupting ecosystems. Interdisciplinary green chemistry approaches aim to design chemicals and processes that are inherently less hazardous, such as using natural colorant extracts for dyeing and coating applications, fiber technologies for improving fiber properties, and enzymatic processes to reduce the chemical footprints at different levels of commercial processing.

## Figures and Tables

**Figure 1 polymers-17-00871-f001:**
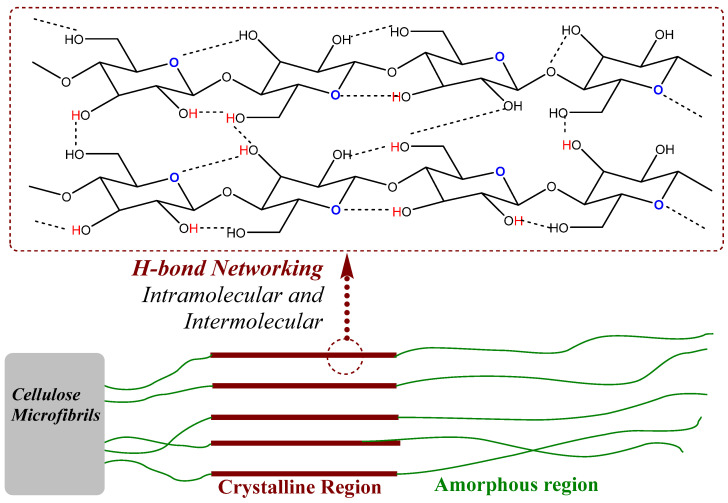
The crystallinity of cellulose materials (fiber) exhibits a high H-bonding networking (intramolecular as well as intermolecular).

**Figure 2 polymers-17-00871-f002:**
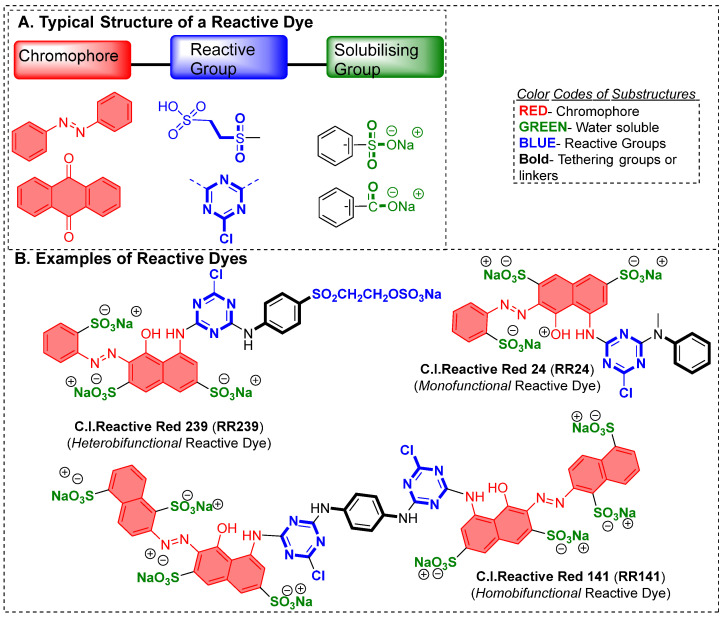
(**A**) Schematic representation of a reactive dye structure: chromophores (in red: Diazo benzene and Anthraquinone); reactive groups (in blue: vinyl sulphone, monochlorotriazine); solubilizing groups (in green: sulphonates and carboxylates); in bold (linkers or tethering groups). Linkers have no functional role but are added to the substructure to link or tether the functional groups into one structure. (**B**) Examples of reactive dyes based on diazo benzene structures (Reactive Red 239, Reactive Red 24, Reactive Red 141).

**Figure 3 polymers-17-00871-f003:**
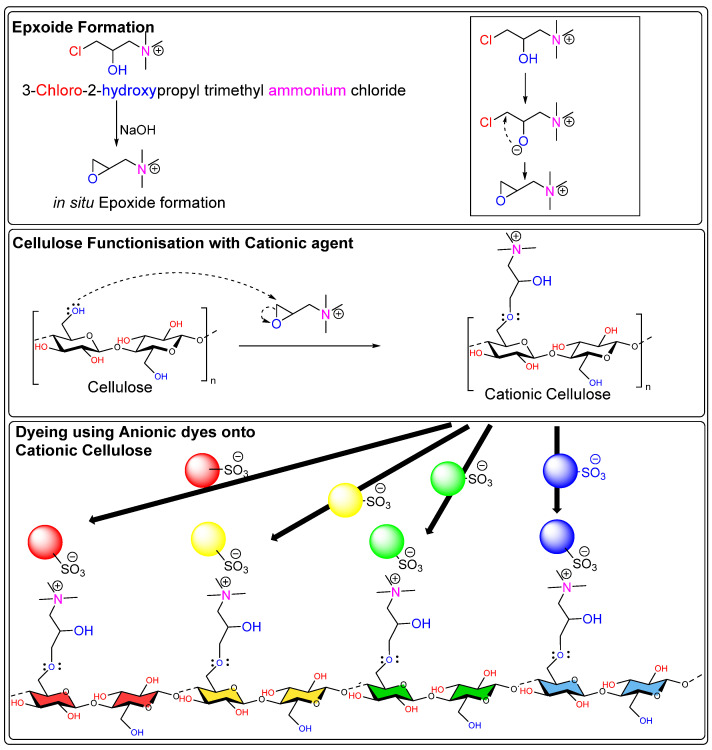
Reaction mechanism of coloration or dyeing of precationized cellulose [89].

**Figure 4 polymers-17-00871-f004:**
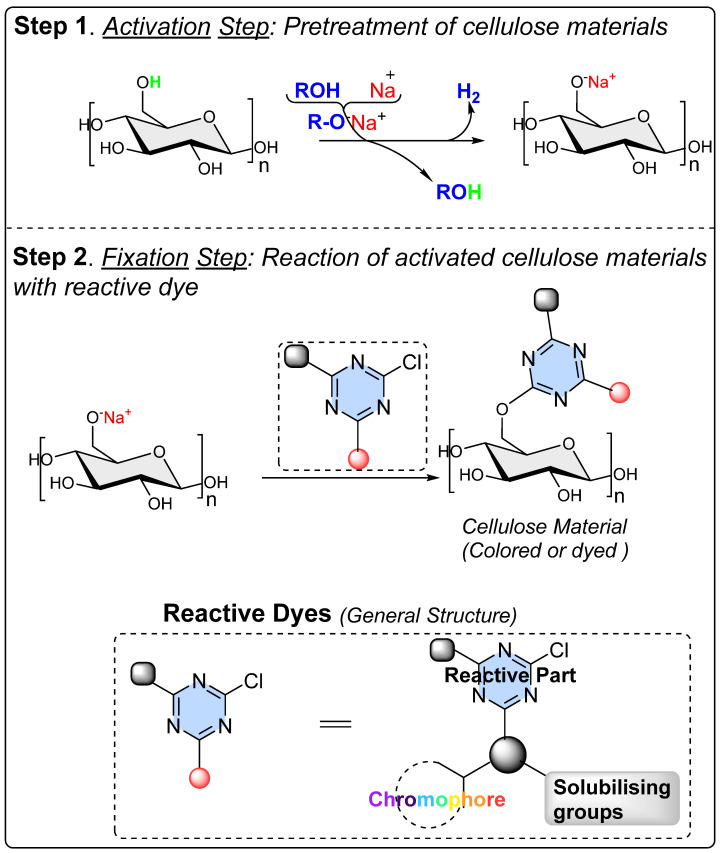
Schematic representation of alkoxide-assisted reactive dyeing (stepwise) for cellulose materials.

**Figure 5 polymers-17-00871-f005:**
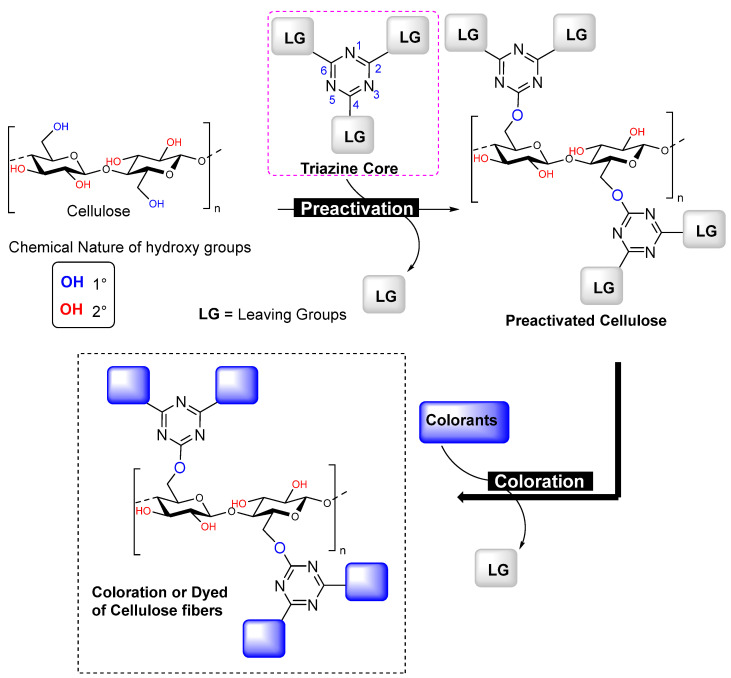
Pretreatment of cellulose materials to enhance the reactivity towards the dye or colorants.

**Figure 6 polymers-17-00871-f006:**
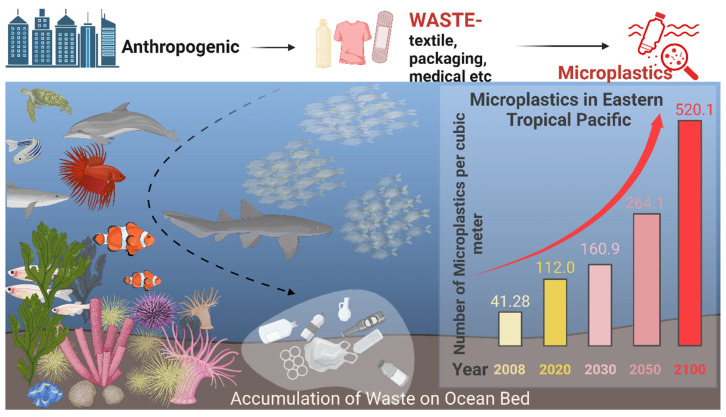
Microplastic emissions: Sources and their environmental impact on our ecosystems.

**Figure 7 polymers-17-00871-f007:**
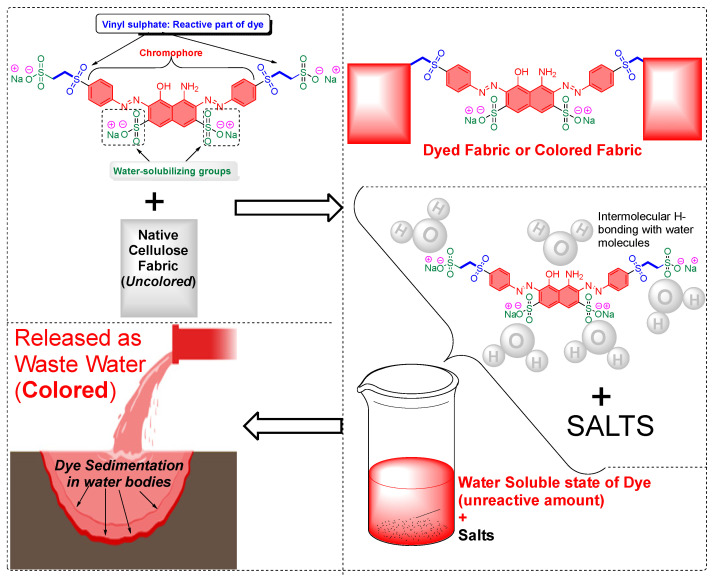
Water pollution from textile wastewater: The presence of water-soluble groups in dye structures (Reactive Black 5 dye structure used as an example here) allows a significant proportion of dye to remain dissolved in water, and requires chemical auxiliaries to enhance dye migration towards fibers. As a result, textile wastewater contains a high concentration of chemical auxiliaries and colorants (dye material), which are ultimately discharged and tend to accumulate on the bottom of water bodies.

**Figure 8 polymers-17-00871-f008:**
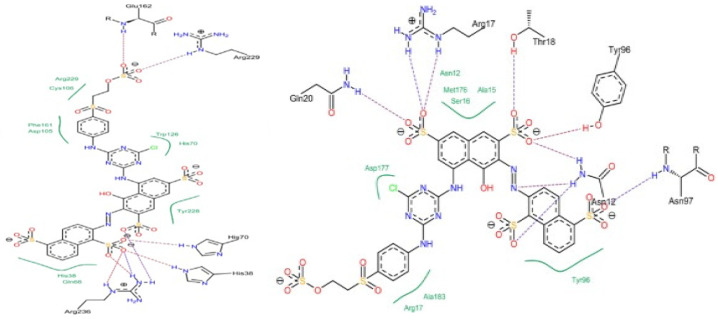
Laccase (LHS) and Azoreductase (RHS) complex with Drimaren Red CL-5B, with *A. hydrophila* MTCC 1739. The figures were reproduced with permission from Srinivasan et al. [182] Copyright 2019 Elsevier.

**Figure 9 polymers-17-00871-f009:**
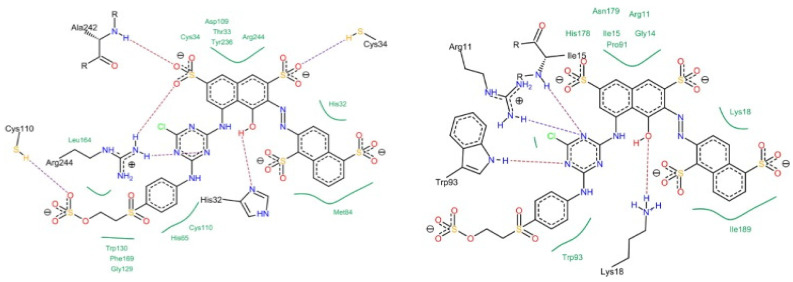
Laccase (LHS) and Azoreductase (RHS) complex with Drimaren Red CL-5B with *L. sphaericus* MTCC 9523. The figures were reproduced with permission from Srinivasan et al. [182] Copyright 2019 Elsevier.

**Table 1 polymers-17-00871-t001:** Classification of synthetic dyes that are used in textile industries.

Textile Dyes						
*Nature*	Non-Ionic		Cationic	Anionic		
	Disperse	Vat or Sulfur dyes		Acid	Direct	Reactive
** *Aqueous* ** ** *solubility* **	Insoluble	Insoluble	Soluble	Soluble	Soluble	Soluble
** *Fiber* ** ** *type* **	Synthetic fibers (polyester)	Cellulose	Wool Acrylic	Wool Silk Nylon	Cellulose	Cellulose
**Color Spectrum ***	Vibrant	Deep, Rich	Bright	Bright, vibrant	Dull	Bright
**Chemical Auxiliaries**	Dispersing agents, Leveling agents, Buffering agents	Reducing agents, Alkaline agents, Oxidising agents	Leveling agents, Acidic agents, Fixatives	Acidic agents, Leveling agents, Salts	Leveling agents, Salts pH adjusters	Salts, Alkaline agents, Fixatives
**Fastness Properties ***	Excellent fastness	Excellent wash fastness	Moderate-to-good fastness	Moderate-to-good fastness	Poor fastness	Excellent fastness
**Examples**	Disperse Yellow 3, Disperse Red 1, Disperse Blue 35, Disperse Blue 56, Disperse Violet 26,	Vat Blue 1, Vat Blue 4, Vat Green 1, Sulfur Black 1, Sulfur Yellow 3,	Cationic Blue 3 Cationic Yellow 2 Cationic Red 2 Cationic Violet 5 Cationic Green 1	Acid Blue 113 Acid Red 27 Acid Yellow 17 Acid Green 25	Direct Blue 1 Direct Red 28 Direct Yellow 86 Direct Green 6	Reactive Blue 239 Reactive Red 180 Reactive Black 5

** depends on the dye-to-dye structure and dyeing conditions.*

## Data Availability

No new data was created or analyzed in this study.

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
