# Peer review of "Environmental Impact of Textile Materials: Challenges in Fiber–Dye Chemistry and Implication of Microbial Biodegradation"

_polymers, 2025, doi:10.3390/polym17070871_

Round 1

Reviewer 1 Report

Comments and Suggestions for Authors

The current review entitled " Environmental Impact of Textile Materials: Challenges in Fiber-Dye Chemistry and Implication of Microbial Biodegradation" explores methods that make use of different microbial systems while analyzing the chemistry, processes, and chemical waste related to the textile industry. This review also discusses microbe integration as a viable way to handle large amounts of textile chemical waste to advance sustainable technology.

Comment:

Although the paper gives a good summary of the topic, it would be helpful to explain more particular case studies or examples to show how the problem is big.
The review’s impact would be increased by include quantitative data on the textile industry's effects on the environment, such as energy and water use and emissions of pollutants.

For fiber chemistry, consider extending the discussion on new sustainable fiber technologies.

Regarding the difference between basic and acid dyes, acid dyes are preferred over basic dyes because they generally offer better lightfastness properties. It might have greater effect if it included a data point or example that demonstrated the variation in lightfastness.

In acidic conditions, the NH2 groups on protein fibers become protonated, attracting the negatively charged acidic dyes with sulfate groups. It would be more thorough to explain how this interaction contributes to increased dyeing stability or efficiency.

For microbial biodegradation of textile dyes, enzymatic activities in addition to aerobic and anaerobic pathways should be detailed discussed.

Types of microorganisms involved in biodegradation of textile dyes should be separately illustrated.

More explanation should be considered for factors affecting biodegradation; such as, temperature, pH, dye structure, concentration, and co-substrates.

Other operational challenges such as toxicity of dyes and by-products, incomplete degradation, and adaptation of microbes should be covered.

Table 1 should be revised and reorganized.

Author Response

The current review entitled " Environmental Impact of Textile Materials: Challenges in Fiber-Dye Chemistry and Implication of Microbial Biodegradation" explores methods that make use of different microbial systems while analyzing the chemistry, processes, and chemical waste related to the textile industry. This review also discusses microbe integration as a viable way to handle large amounts of textile chemical waste to advance sustainable technology.

Although the paper gives a good summary of the topic, it would be helpful to explain more particular case studies or examples to show how the problem is big.
The review’s impact would be increased by include quantitative data on the textile industry's effects on the environment, such as energy and water use and emissions of pollutants.

Author Response: The author would like to express gratitude to the reviewer for their thoughtful comments and for taking the time to provide feedback. To address the reviewer’s comments, we have included continuous line numbering and enabled track changes, highlighting modifications in yellow/other colors for the reviewer’s or editor's convenience.

(1) For fiber chemistry, consider extending the discussion on new sustainable fiber technologies.

Author Response: A new section on sustainable fiber technology has been added, highlighting recent advancements in Ioncell technology, Tencel technology, and waterless processing methods.

(2) Regarding the difference between basic and acid dyes, acid dyes are preferred over basic dyes because they generally offer better lightfastness properties. It might have greater effect if it included a data point or example that demonstrated the variation in lightfastness.

Author Response: The section has been rephrased, and a brief rationale has been included.

(3) In acidic conditions, the NH2 groups on protein fibers become protonated, attracting the negatively charged acidic dyes with sulfate groups. It would be more thorough to explain how this interaction contributes to increased dyeing stability or efficiency.

Author Response: Additional information has been included in the protein fiber dyeing section. Please review the updates in the file with tracked changes.

(4) For microbial biodegradation of textile dyes, enzymatic activities in addition to aerobic and anaerobic pathways should be detailed discussed.

Author Response: A section dedicated to this information has been added, highlighted in yellow.

(5) Types of microorganisms involved in biodegradation of textile dyes should be separately illustrated.

Author Response: This information is presented in different sections of the paper. However, our study is limited to research that supports the rationale behind how certain microorganisms outperform others in the degradation of specific dyes.

(6) More explanation should be considered for factors affecting biodegradation; such as, temperature, pH, dye structure, concentration, and co-substrates.

Author Response: The information is revised and varous sections of the paper contain this information, with the newer additions marked in cyan.

(7) Other operational challenges such as toxicity of dyes and by-products, incomplete degradation, and adaptation of microbes should be covered.

Author Response: This information is rewritten in different sections of the paper.

(8) Table 1 should be revised and reorganized.

Author Response: Some alterations have been made to Table 1. Please refer to the Track Changes file for more details.

Reviewer 2 Report

Comments and Suggestions for Authors

The manuscript contains a comprehensive review on types and sources of textiles materials, the merits and challenges of the production process to man and his environment. It will be of interest to the scientific community, if the author can resolve the following identified issues:

1. The use of English must be improved throughout the manuscript.

2. Some of the claims made by the author are incorrect, for instance, the claim of viscous rayon as natural fibers,the claim of addition of salt leading to hydrolysis of Azo dye etc

3. All acronyms must be defined at first mention.

4. Lack of enough references in the Introduction section. Citation of the following references will boost the quality of this manuscript: Current Research in Green and Sustainable Chemistry 4 (2021) 100151; International Journal of Sustainable Engineering 17 (2024) 1-10; Science Progress 107(3) (2024) 1-22. Author is encouraged to cite more recent journals.

5. Section 2.1.1: Citation of the following references can enhance the readership of this manuscript: Arabia Journal of Chemistry 13(5) (2020) 5417-5429; Journal of Cleaner Production 267 (2020) 121903

6. Author's claim in line 272-274 is wrong. The parameters for acid dyeing of fibers are pH, nature of fiber, initial dye concentration, temperature and liquor ratio.

7. Line 303: non ionic? polycotton? What are the meaning of these?

8. Some chemical formulae are wrongly written by the author, like Na2CO3 in line 417.

8. Many of the references are poorly cited, like Negi et al. in line 438. It has no reference number.

9. Some sentences are too long, author is advised to break them. For instance, line 573-579, among others.

Comments on the Quality of English Language

The use of English needs to be enhanced throughout the manuscript. Typo and grammatical errors.

Author Response

The manuscript contains a comprehensive review on types and sources of textiles materials, the merits and challenges of the production process to man and his environment. It will be of interest to the scientific community, if the author can resolve the following identified issues:

Author Response: The author would like to express gratitude to the reviewer for their thoughtful comments and for taking the time to provide feedback. To address the reviewer’s comments, we have included continuous line numbering and enabled track changes, highlighting modifications in yellow/other colors for the reviewer’s or editor's convenience.

  1. The use of English must be improved throughout the manuscript.

Author response: Almost all sections are rewritten to an extent, please follow the file that has the track changes ON option.

  1. Some of the claims made by the author are incorrect, for instance, the claim of viscous rayon as natural fibers,the claim of addition of salt leading to hydrolysis of Azo dye etc

Author response: Thanks for pointing out the information, as it is now accordingly revised.

  1. All acronyms must be defined at first mention.

Author response: There are changes made according to the manuscript.

  1. Lack of enough references in the Introduction section. Citation of the following references will boost the quality of this manuscript: Current Research in Green and Sustainable Chemistry 4 (2021) 100151; International Journal of Sustainable Engineering 17 (2024) 1-10; Science Progress 107(3) (2024) 1-22. Author is encouraged to cite more recent journals.

Author response: Thanks for the suggestions, additions were made to the manuscript.

  1. Section 2.1.1: Citation of the following references can enhance the readership of this manuscript: Arabia Journal of Chemistry 13(5) (2020) 5417-5429; Journal of Cleaner Production 267 (2020) 121903

Author response: Thanks for the suggestions, additions were made to the manuscript.

  1. Author's claim in line 272-274 is wrong. The parameters for acid dyeing of fibers are pH, nature of fiber, initial dye concentration, temperature and liquor ratio.

Author response: Thanks for the suggestions, detailing in the running text is modified accordingly.

  1. Line 303: non ionic? polycotton? What are the meaning of these?

Author Response: The sentence is modified as, “The application of reactive dyes in the 1950s for cellulose material coloration (dyeing) was identified, where the reactive dyes in an alkaline pH were found to form non-ionic covalent bonds with the hydroxy groups of glucose units (cellulose fibers).”

  1. Some chemical formulae are wrongly written by the author, like Na2CO3 in line 417.

Author Response: It is written now as Na2CO3.

  1. Many of the references are poorly cited, like Negi et al. in line 438. It has no reference number.

Author Response: Citation is added according to the reviewer's suggestion.

  1. Some sentences are too long, author is advised to break them. For instance, line 573-579, among others.

Author Response: The use of English is enhanced throughout the manuscript. Typo and grammatical errors are rectified.

Round 2

Reviewer 2 Report

Comments and Suggestions for Authors

Accept